**Molecular compositions and optical properties of dissolved brown carbon in biomass burning, coal combustion, vehicle emission aerosols illuminated by excitation-emission matrix spectroscopy and FT-ICR MS analysis**

Jiao Tang[1,4], Jun Li[1], Tao Su[1,4], Yong Han[2], Yangzhi Mo[1], Hongxing, Jiang[1,4], Min Cui[2, a], Bin Jiang[1], Yingjun Chen[2], Jianhui Tang[3], Jianzhong Song[1], Ping'an Peng[1], Gan Zhang [1]

[1]State Key Laboratory of Organic Geochemistry and Guangdong Key Laboratory of Environmental Protection and Resources Utilization, Guangzhou Institute of Geochemistry, Chinese Academy of Sciences, Guangzhou 510640, China

[2]Department of Environmental Science and Engineering, Fudan University, Shanghai 200092, P.R. China

[3]Key Laboratory of Coastal Environmental Processes and Ecological Remediation, Yantai Institute of Coastal Zone Research, Chinese Academy of Sciences, Yantai 264003, China

[4]University of Chinese Academy of Sciences, Beijing 100049, China

[a]now at: School of Environmental Science and Engineering, Yangzhou University, Yangzhou 225127, China

**Correspondence**: Jun Li (junli@gig.ac.cn) and Gan Zhang (zhanggan@gig.ac.cn)

**Abstract:** Brown carbon (BrC) plays an essential impact on radiative forcing due to its ability to absorb sunlight. In this study, the optical properties and molecular characteristics of water-soluble and methanol-soluble organic carbon (MSOC) emitted from the simulated combustion of biomass and coal fuels, and vehicle emissions were investigated using UV-visible spectroscopy, excitation-emission matrix (EEM) spectroscopy, and Fourier-transform ion cyclotron resonance mass spectrometry (FT-ICR MS) coupled with electrospray ionization (ESI). The results showed that these smoke aerosol samples from biomass burning (BB) and coal combustion (CC) had a higher mass absorption efficiency at 365 nm ($MAE_{365}$) than vehicle emission samples. A stronger $MAE_{365}$ value was also found in MSOC than water-soluble organic carbon (WSOC), indicating low polar compounds would possess higher light absorption capacity. Parallel factor analysis (PARAFAC) identified six types of fluorophores (P1−6) in WSOC including two humic-like substances (HULIS-1) (P1, and P6), three protein-like substances (PLOM) (P2, P3, and P5), and one undefined (P4). HULIS-1 was mainly from aging vehicle-exhaust particles, P2 was only abundant in BB aerosols, P3 was ubiquitous in all tested aerosols, P4 was abundant in fossil burning aerosols, and P5 was more intense in fresh vehicle-exhaust particles. The MSOC chromophores (six components, C1−6) exhibited consistent characteristics with WSOC, suggesting the method could be used to indicate the origins of chromophores. FI-ICR mass spectra showed that CHO and CHON were the most abundant components of WSOC, but S-containing compounds appeared a higher abundance in CC aerosols and vehicle emission than BB aerosols. While, considerably low S-containing compounds with largely CHO and CHON were detected in MSOC. The unique formulas of different sources determined by the Venn diagram presented different molecular distributions. To be specific, BB aerosols with largely CHO and CHON had a medium H/C and low O/C ratio; while, CC aerosols and vehicle emissions with largely S-containing compounds had an opposite H/C and O/C ratio. Moreover, the light absorption capacity of WSOC and MSOC was positively associated with the unsaturation degree and molecular weight in the source aerosols. The above results are potentially applicable to further studies on EEM-based

or molecular characteristic-based source apportionment of chromophores in

atmospheric aerosols.

## 1 Introduction

Carbonaceous aerosols play an important role in the Earth's radiative balance. One such aerosol, black carbon (BC), absorbs significant amounts of light and exerts a warming effect, while organic carbon (OC) was initially thought to only scatter solar radiation (Wong et al., 2017;Mo et al., 2017;Saleh et al., 2014). However, recent studies show that there are certain types of OC that absorb radiation efficiently in the near-ultraviolet (UV) (300–400 nm) and visible ranges, which are called brown carbon (BrC). They can positively shift the net direct radiation forcing (DRF) (Saleh et al., 2014;Laskin et al., 2015;Kirchstetter and Thatcher, 2012). According to a simulation model, the inclusion of BrC may enhance total aerosol absorption by 7–19% (Feng et al., 2013). According to previous studies, BrC in atmospheric aerosols mainly originates from emissions from biomass burning (BB) and coal combustion (CC), vehicle exhausts, and the formation of secondary organic aerosol (SOA) (Zhu et al., 2018;Laskin et al., 2015;Xie et al., 2017;Kumar et al., 2018). Among them, primary emissions contribute significantly to BrC absorption (Fan et al., 2012;Yan et al., 2015;Zhang et al., 2011). Recently, many studies have investigated the optical properties and molecular characteristics of BrC in laboratory simulated combustion (Budisulistiorini et al., 2017;Lin et al., 2018;Lin et al., 2016;Song et al., 2019) and their light absorption in controlled vehicle emissions (Xie et al., 2017). However, there were no available studies on the comprehensive characteristics of BrC in various sources and their variations in optical and chemical information impacted by these sources. Therefore, investigating BrC in different sources would improve our understanding of the evolution of BrC absorption.

Excitation-emission matrix (EEM) spectroscopy can provide structure information of chromophores and thus has been widely applied to identify the sources and chemical nature of chromophoric dissolved organic matter (CDOM) in aquatic environments since the 1990s (Shimabuku et al., 2017;Wells et al., 2017;Bhattacharya and Osburn, 2017;Coble, 1996). Due to the optical properties of chromophoric water-soluble organic carbon (WSOC) in the atmosphere were similar to CDOM in aquatic environments (Qin et al., 2018;Fu et al., 2015;Graber and Rudich, 2006), this

technique could extend to atmospheric research. It has to be mentioned that
fluorescence is a radiative process that occurs between two energy levels of the same
multiplicity (Andrade-Eiroa et al., 2013). Generally, compounds with rigid planar
structures and highly conjugated systems have intrinsic fluorescence emission
characteristics and are important BrC chromophores, such as aromatic acids, phenols,
nitroaromatics, polycyclic aromatic hydrocarbons (PAHs), quinones, and so on (Lin et
al., 2018;Zhang et al., 2013). In addition, chromophores in fluorescence spectra could
be considered as a "fingerprinting" tool, especially when combining it with parallel
factor (PARAFAC) analysis which can decompose EEM signals into their underlying
chemical components (Murphy et al., 2013). For instance, Chen et al. (2016b)
observed that the water-extracted chromophores identified by PARAFAC from the
urban, forest, and marine aerosols were varied with the sampling sites and periods,
and were affected by oxidative and functional groups. Lee et al. (2013) illustrated that
SOA derived from the oxidation of limonene and decene with $O_3$ and OH had
different fluorescence spectra. Therefore, BrC characteristics from various sources
may differ. However, when analyzing chromophoric BrC using fluorescence spectra,
the challenges are the lack of a classification system for fluorescence spectra, to
distinguish chromophores from most non-absorbing constituents and to determine the
chemical structures of the chromophores.
Fourier-transform ion cyclotron resonance mass spectrometry (FT-ICR MS)
coupled with electrospray ionization (ESI) is a powerful platform for investigating the
detailed characteristics of organic material at the molecular level. With the advantage
of ultrahigh-resolution, the accuracy of mass measurements, and high sensitivity
(Feng et al., 2016), FT-ICR MS has been successfully used to characterize organic
aerosols (Jiang et al., 2016;Song et al., 2018;Mo et al., 2018), cloud water (Zhao et al.,
2013), and natural organic matters (Sleighter et al., 2012;Feng et al., 2016). For
example, a previous study has determined their molecular families of dissolved
organic matters (DOMs) associated with fluorescent components by using FI-ICR MS
(Stubbins et al., 2014), which could provide more chemical information of
chromophores.
Residential CC and BB emissions, and motor vehicle emissions are significant
anthropogenic sources of air pollutants, exceptionally fine particulate matter ($PM_{2.5}$)
on urban and regional scales (Gentner et al., 2017;Yan et al., 2015;Zhang et al.,
2018;Chen et al., 2015). In this study, to obtain a comprehensive understanding of
BrC originating from various sources, UV-vis, EEM, and FI-ICR MS analysis were
performed for WSOC and methanol-soluble organic carbon (MSOC) from the smoke
particles of simulated combustion of biomass fuels and coals, and vehicle emission
aerosols. Statistical analysis of PARAFAC was applied to EEM spectra to resolve the
fluorescent compounds. All, and unique molecular characteristic of WSOC and
MSOC were analyzed and discussed on the base of FI-ICR MS. Relationships
between optical properties and chemical structures were discussed by using linear
regression coefficient.
**2 Experimental methods**
**2.1 Sample collection and preparation**
The smoke particles were collected by the instrument coupled with a dilution channel
which was designed to simulate fire emissions representative of "real-world" open BB
and household CC activities (Figure S1). In the present study, a total of 27 BB
samples (IDs1−27) were collected at Xishuangbanna city, Yunnan Provence, from
May 20th to June 3th, 2016 and the detailed sampling process was described in our
previous article (Cui et al., 2018). Briefly, raw fuels (rough $20\times3\times2$ cm$^3$) were
air-dried for several days and ignited in a stainless-steel bowl, and then the rising
smoke was collected through a dilution system. The sampling system mainly consists
of a dilution tunnel, a residence time chamber, three particulate matter (PM) samplers,
and so on. Every biomass was burned three times, about 1−2 kg fuels per burn. Every
combustion process lasted for 20 minutes. The collection of smoke particles started
when the fuel ignited, and ended when the concentration of $CO_2$ down to atmosphere
$CO_2$ level. Dilution ratios of each experimental process were calculated using the $CO_2$
concentrations before and after dilution. The collection flow rate and average dilution
ratio were 180 L/min and 2.1, respectively. And the other 6 BB samples (IDs28−33)
were collected in Guangzhou city, Guangdong Province.
The smoke particles of CC (IDs34−50) were collected as same as that of BB
experiment, but used a stove, in Guangzhou city, Guangdong province, from
November 18th, 2017 to January 23th, 2018. The tested stove is technically improved
stove (named Jin-Yin stove). Due to the difficulty of ignition of coal, we used
smokeless charcoal to ignite one-third (about 300 g) of the raw-coal chunk (2−5 cm in
size) in the stove, removed the charcoal after ignition, and then added the remaining
raw-coal chunk (about 700 g) to start to collect the smoke particles. Every coal was
also burned three times, about 1 kg fuels per burn. Every combustion process lasted
for about 40−150 minutes. The collection flow rate and average dilution ratio were
150 L/min and 1.5, respectively. Additionally, modified combustion efficiency (MCE)
was calculated to characterize the relative amount of smoldering and flaming
combustion phase (Lin et al., 2016;Cui et al., 2018). The average MCE value was
0.73 ± 0.08 for CC experiments but unavailable for the BB experiments because the
CO sensor did not work in the field work, which was mentioned in our previous paper
(Cui et al., 2018).
Tunnel aerosols (total eight samples, IDs51−58) were collected at Siping Tunnel
from November 1th to 2th, 2017 and Xiaoyangshan Tunnel from December 1th to 2th,
2017, in Shanghai city, as well as two vehicle exhaust particles (IDs59−60) were
collected from the direct emission of two different trucks (more fresh aerosols). With
no other instructions, we used "vehicle emissions" to represent all tunnel aerosols and
vehicle exhaust particle samples. These filters were wrapped in aluminum foil and
pre-baked at 450 °C for 5 hours before sampling and stored at −20 °C after sampling.
Overall, there was a total of 60 total suspended particulate matter (TSP) samples on
source emissions in this study, and blank samples that were collected at different
times and locations were used for correcting filter samples.
WSOC for UV-Vis absorption and EEM analysis was extracted with purified
water (resistivity of >18.2Ω) via ultra-sonication of quartz filter punches for 30
minutes. Because water cannot effectively extract the BrC (Liu et al., 2013;Shetty et

al., 2019), the remaining filter was further freeze-dried and extracted with methanol (HPLC grade) to obtain the MSOC constituent for better understand the optical properties and molecular compositions of BrC. It is worth noting that the MSOC in this study is not necessarily like that of the same names in other studies. All the extracts were filtered through a 0.22 μm polytetrafluoroethylene membrane into amber colored glass vials to remove the insoluble material.

**2.2 Carbon analysis**

We measured both OC and elemental carbon (EC) using an aerosol carbon analyzer (Sunset Laboratory, Inc., USA), following the NIOSH thermal-optical transmittance (TOT) standard method (Mo et al., 2017), and the emission factors (EFs) of PM, OC and EC were calculated and detail information was presented in Supplement. We also analyzed the elements of biomass (C, H, O, and N) and coal (C, H, O, N, and S) using an elemental analyzer (Vario EL cube; Elementar, Germany) and the results were listed in Table S1 and S2. The carbon content of WSOC was measured using total organic carbon analysis (Vario TOC cube; Elementar) before acidifying with phosphoric acid to remove inorganic carbon, while the concentration of MSOC was assessed using the method developed by a previous study (Chen et al., 2017b). Briefly, the extracted MSOC was dried gently under nitrogen, and then re-dissolved in 500 µL methanol. Subsequently, 50 µL of the solution was added to the clear quartz filter (area: 1.5 cm$^2$) until dry and analyzed using the TOT standard method.

**2.3 UV−Vis absorption spectra and EEM fluorescence spectra**

The UV-vis absorption and EEM spectra of WSOC and MSOC were analyzed using a UV-Vis spectrophotometer (UV-4802; Unico, China) and an Aqualog fluorometer (Horiba Scientific, USA), respectively. The wavelengths used to characterize the UV-vis spectra were between 200 to 800 nm at a step size of 2 nm. Purified water and methanol were used as a baseline correction for WSOC and MSOC before measure, respectively. Mass absorption efficiency (MAE, m$^2$ g$^{-1}$ C) was obtained as the following equation (Li et al., 2018):

$MAE_\lambda = A_\lambda \bullet \ln(10) / (C \bullet L)$            (1)
Here, $A_\lambda$ is the value of light absorption at the given wavelength of the
spectrophotometer; $C$ ($\mu g$ C mL$^{-1}$) is the concentration of WSOC and MSOC; $L$ is the
optical path length (in this study, 0.01m). Moreover, the pH of WSOC was measured
for all samples within the ranges of 5.5−6.5, which generally did not affect the
absorbance according to the prior study (Chen et al., 2016a).
The emission and excitation wavelengths of the fluorescence spectra were from
245 to 580 nm and 240 to 500 nm, respectively. The wavelength increments of the
emission and excitation scans were 4.66 and 3 nm, respectively. Further, the
contributions of solvents to the fluorescence spectra were subtracted.
**2.4 Ultrahigh-resolution ESI FT-ICR MS analysis**
The WSOC and MSOC of six selected samples including two BB aerosols (Musa and
Hevea), two CC aerosols (a anthracite and a bituminous coal), one day of tunnel
aerosol (combine the aerosols in inlet and outlet of the tunnel in the same day), and
one vehicle exhaust particle were analyzed using FT-ICR MS. To remove inorganic
ions before instrumental analysis, WSOC was further adjusted to pH = 2 by the
addition of hydrochloric acid (HCl) and then passed through a solid-phase extraction
cartridge (Oasis HLB, 30 um, 60 mg/cartridge; Waters Corporation, USA). The
constituent retained on the SPE cartridge was eluted with methanol containing 2%
ammonia (v/v). Eluants were evaporated until dry under a gentle nitrogen gas stream.
The extracted solutions by methanol was evaporated under a gentle nitrogen gas
stream for preparation.
We used the analysis method of FT-ICR MS described in detail in our previous
study (Mo et al., 2018). Briefly, ultrahigh-resolution mass spectra were obtained using
a solariX XR FT-ICR MS (Bruker Daltonics GmbH, Bremen, Germany) equipped
with a 9.4-T superconducting magnet and an ESI ion source. The system was operated
in negative ionization mode. The ion accumulation time was set to 0.6 s. The lower
and upper mass limit was set to m/z 150 and 800 Da, respectively. The mass spectra
were externally calibrated with arginine clusters using a linear calibration and then
internally recalibrated with typical $O_6S_1$ class species peaks using quadratic
calibration in DataAnalysis ver. 4.4 software (Bruker Daltonics). A typical
mass-resolving power >450 000 at m/z 319 with <0.2 ppm absolute mass error was
achieved. The mass spectra of field blank filters was analyzed to detect possible
contamination following the same procedures. More data processing was presented in
S1 of the Supplement.
**2.5 PARAFAC analysis for EEM spectra**
PARAFAC analysis with non-negativity constraints was used to explore the
fluorescent components in dissolved BrC based on the method established by the
previous studies (Murphy et al., 2013;Andersson and Bro, 2000), which was
performed using drEEM toolbox version 2.0 using a MATLAB software
(http://models.life.ku.dk/drEEM). This method had been widely used in the analysis
of fluorescence spectra in aerosols (Chen et al., 2016b;Chen et al., 2016a;Matos et al.,
2015;Wu et al., 2019). Absorbance measurements were used to correct the EEM for
inner filter effects (IFE) according to the previous studies (Luciani et al., 2009;Gu and
Kenny, 2009;Fu et al., 2015). The highest light absorbance in the calibrated
wavelength range of WSOC and MSOC was not greater than 2 (mostly below 1 at 254
nm), which was appropriate for the inner filter corrections of the EEMs (Gu and
Kenny, 2009;Murphy et al., 2013). Each EEM was normalized to the Raman peak
area of purified water collected on the same day to correct fluorescence in Raman
Units (RU) at excitation 350 nm and corrected for the dilution factor (Murphy et al.,
2013;Murphy et al., 2010). Additionally, the signals of the first-order and
second-order Rayleigh and Raman scattering in the EEM were removed by an
interpolation method (Bahram et al., 2006). Repeated convergence of the model was
examined based on the iteration of the minimum square principle. The exploration
phases of 2- to 7-components PARAFAC models contained an evaluation of the shape
of spectral loading, leverage analysis, an examination of the core consistency, residual
analysis, and split-half analysis (Figure S2−S7). Six-component PARAFAC model
was identified and successfully passed the split-half validation with the split style of
"$S_4C_6T_3$" for the WSOC and MSOC in 60 samples, respectively.
**3 Results and discussions**
**3.1 Emission characteristics and light absorption of Extracts**
The PM, OC, and EC EFs of 27 biomass and 17 coal combustion experiments were
summarized in Table S3. The relevant EFs of some biomass species have been
previously reported (Cui et al., 2018). In this experiment, the EFs of PM, OC, and EC
from 27 types of biomass burning were $15 \pm 11$ g kg$^{-1}$ fuel, $8.0 \pm 6.4$ g kg$^{-1}$ fuel, and
$7.7 \times 10^{-1} \pm 3.4 \times 10^{-1}$ g kg$^{-1}$ fuel, respectively. The EFs emitted from bituminous CC
(PM $= 9.1 \times 10^{-1} \pm 6.5 \times 10^{-1}$ g kg$^{-1}$ fuel, OC $= 4.2 \times 10^{-1} \pm 3.3 \times 10^{-1}$ g kg$^{-1}$ fuel, EC $=$
$9.4 \times 10^{-2} \pm 1.9 \times 10^{-1}$ g kg$^{-1}$ fuel) were much higher than those of anthracite
combustion (PM $= 1.5 \times 10^{-1} \pm 8.9 \times 10^{-2}$ g kg$^{-1}$ fuel, OC $= 1.2 \times 10^{-2} \pm 4.5 \times 10^{-3}$ g
kg$^{-1}$ fuel, EC $= 1.6 \times 10^{-4} \pm 1.4 \times 10^{-4}$ g kg$^{-1}$ fuel) in the same stove. These differences
could be attributed to the high volatile matter content of bituminous coal (Tian et al.,
2017;Chen et al., 2005). Note that the CC smoke collection began when the flame had
been ignited with one-third of the material and the rest was added. Therefore, the
results of our study would be lower than the real values.
MAE can be used to characterize the efficiency of solar energy absorption, which
is represented by the degree of conjugation and the amount of electron delocalization
in molecules (Chen et al., 2016a). As shown in Figure 1 and Table S4, MAE at 365
nm (MAE$_{365}$) was significantly higher in the case of BB and CC aerosols than in
vehicle emissions in this study, consistent with the previous findings (Xie et al.,
2017;Fan et al., 2016). Bituminous CC aerosols had higher MAE$_{365}$ values than
anthracite combustion aerosols. Here, we introduced the EC/OC ratios, which could
be used as an indicator of fire conditions (Xie et al., 2017). Figure S8 showed the
MAE$_{365}$ of WSOC vs. EC/OC relationships for all BB and CC aerosols. The data
clearly showed that the WSOC light absorption of BB aerosols was dependent on
combustion conditions. However, weak relationship ($p > 0.05$) in CC aerosols
suggested another factor might influence the light absorption, such as maturity (Li et
al., 2018). Compared to WSOC, higher $MAE_{365}$ values were observed in the MSOC
collected from BB (2.3 ± 1.1 $m^2$ $g^{-1}$C) and bituminous CC (3.2 ± 1.1 $m^2$ $g^{-1}$C) aerosols.
This could be due to the fact that these strongly light-absorbing fat-soluble
components are likely to be large molecular weight PAHs, and quinones from BB and
fossil fuel combustion (Sun et al., 2007;Chen and Bond, 2010), which were more
soluble in low-polarity solution, but we obtained the opposite results in the case of
anthracite combustion and vehicle emissions.
The $MAE_{365}$ of WSOC in this study was compared with the other studies (Figure
1). The BB aerosols in this study had a higher $MAE_{365}$ value than those in other
controlled BB experiments, while it was comparable to corn straw burning emissions
(Park and Yu, 2016;Fan et al., 2016). Further, the simulated BB aerosols exhibited a
higher $MAE_{365}$ value than those in highly BB-impacted areas (Hecobian et al., 2010),
indicating the aging in the transport process could reduce the light absorption (Dasari
et al., 2019). The CC aerosols showed a higher $MAE_{365}$ value than the other coal
experiments (Li et al., 2018;Fan et al., 2016), while comparable values to
water-soluble BrC were observed in winter of Beijing (Cheng et al., 2011;Yan et al.,
2015). The result indicated the strong influence of BrC in this season in this region. In
addition, the simulated combustion aerosols in this study exhibited higher $MAE_{365}$
values than the other areas (such as Guangzhou, Nanjing, Los Angeles, Korea, Nepal,
and so on) (see Figure 1).
Methanol has a lower polarity than water and can extract the water-insoluble
compounds that are generally stronger chromophores. Chen et al. (2017b) extracted
organic matters in aerosols using different polar solutions, and they found
water-insoluble organic matters (WIOM) had a higher MAE value than the
water-soluble organic matters (WSOM), consistent with our result in the BB and
bituminous CC aerosols. Vehicle emission aerosols generally had a lower MAE value
such as methanol-soluble BrC (0.62 ± 0.76 $m^2$ $g^{-1}$C) in the controlled emission
experiment (Xie et al., 2017), which was comparable to WSOC (0.71 ± 0.30 $m^2$ $g^{-1}$C)
but higher than MSOC (0.26 ± 0.09 $m^2$ $g^{-1}$C) in this study.

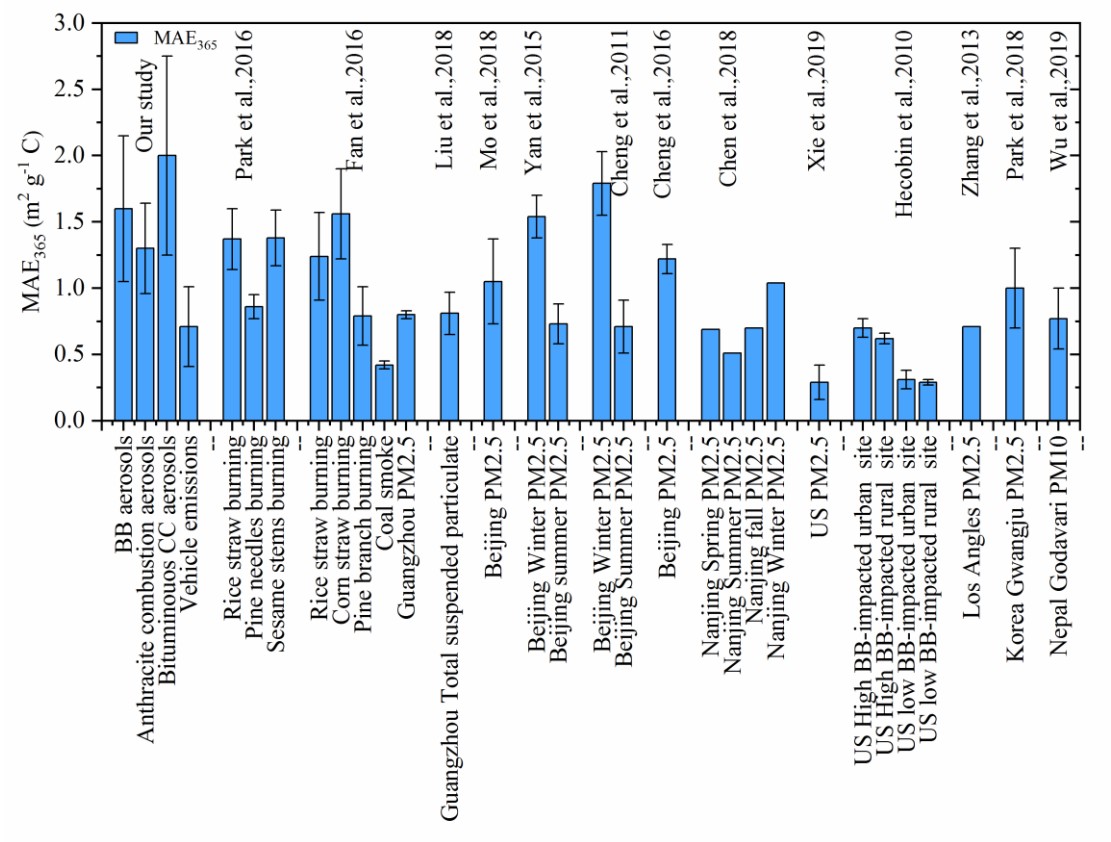


**Figure 1.** Comparison of MAE$_{365}$ in the WSOC fraction of source emission aerosols with the other studies. The references were listed as follows:(Liu et al., 2018;Mo et al., 2018;Yan et al., 2015;Cheng et al., 2011;Cheng et al., 2016;Xie et al., 2019;Hecobian et al., 2010;Zhang et al., 2013;Park et al., 2018;Wu et al., 2019;Fan et al., 2016;Park and Yu, 2016;Chen et al., 2018).

### 3.2 EEM spectra of WSOC and MSOC

Fluorescence spectra was used to characterize the organic chromophores of different sources. We applied the PARAFAC model (Murphy et al., 2013) to determine the underlying chromophore components of the 60 source samples. Six typically independent components (abbreviated P1–6) in WSOC were resolved, as shown in the top of Figure 2 and Table 1. Compared to the previous studies, the fluorescence of P1 and P6 were similar to those of 7CM-C1 (the C1 component of a seven-component model) and 7CM-C3, named humic-like substances (HULIS-1) (Chen et al., 2016b). Further, there were peaks in the emission wavelengths (> 400 nm) of P1 and P6, which were probably derived from conjugated systems (Chen et al., 2016b). The peak of P3 component was mostly located in the region categorized as protein-like

(cytidine) or tryptophan-like fluorophore (Qin et al., 2018;Fan et al., 2016). Generally,
peaks at shorter excitation wavelengths (< 250 nm) and shorter emission wavelengths
(< 350 nm) were associated with simple aromatic proteins such as tyrosine (Cory and
Mcknight, 2005), which was similar to the fluorescence of P2 component observed in
this study. P5 component was similar to tryptophan- and tyrosine-like components
(Chen et al., 2017a). Therefore, P2, P3, and P5 components were named protein-like
substances (PLOM). P4 component was reported relatively rarely but similar to
previously observed peaks that were considered to arise mainly in surface water and
algal secretions (Yu et al., 2015). It is worth noting that the origins and chemical
structures of the chromophores studied are not necessarily like those of chromophores
with the same names in other types of organic matter.
**Table 1.** The maximum excitation and emission wavelengths of the PARAFAC components
in the WSOC and MSOC extracted from the three origins

| | PARAFAC component | Excitation maxima (nm) | Emission maxima (nm) | Assignment according to published papers | References |
|---|---|---|---|---|---|
| WSOC | P1 | 251, 314 | 415 | HULIS-1, terrestrial humic-like component | (Chen et al., 2016b;Sgroi et al., 2017;Fu et al., 2015) |
| | P2 | 254 | 337 | Tyrosine-like | (Cory and Mcknight, 2005) |
| | P3 | 287 | 360 | Protein-like (cytidine) or tryptophan-like | (Qin et al., 2018;Fan et al., 2016) |
| | P4 | 251 | 374 | - | - |
| | P5 | 278 | 319 | Protein-like fluorophores | (Fu et al., 2015) |
| | P6 | 254, 371 | 485 | HULIS-1, conjugated systems, a terrestrial humic or fulvic acid-like component | (Chen et al., 2016b) |
| MSOC | C1 | 308 | 356 | - | - |
| | C2 | <250,272 | 388 | - | - |
| | C3 | <250 | 434 | Component 2 for the urban | (Matos et al., 2015) |

| | | | | |
|---|---|---|---|---|
| C4 | 257 | 360 | - | - |
| C5 | 284 | 328 | - | - |
| C6 | 269 | 310 | - | - |

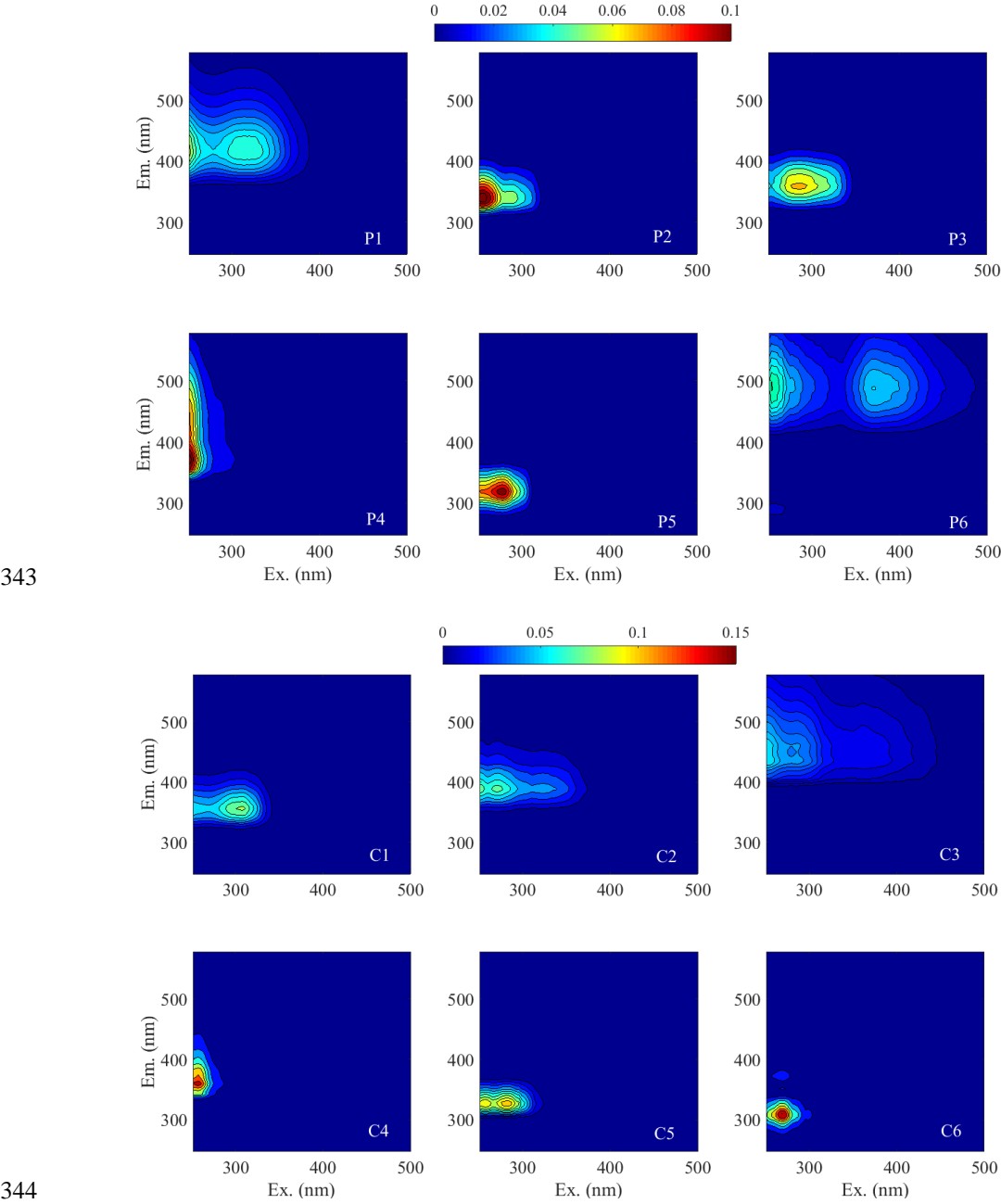



**Figure 2.** The EEM components identified by PARAFAC of WSOC (top: P1−6) and MSOC

(bottom: C1−6) from the three origins.

The results from the six-component model (abbreviated C1–6) of MSOC

identified by PARAFAC, as described in the bottom of Table 1 and Figure 2, were
different from those observed in WSOC, indicating MSOC contained different
compound types from WSOC after water extraction. The peak of C1 component was
similar to that of P3 component of WSOC, but the excitation wavelength was higher
than that of P3 component. The higher excitation wavelength indicated the presence
of conjugated unsaturated bond systems shifting towards the high wavelengths of C1
component (Matos et al., 2015). Moreover, as reported, C3 component was similar to
component 2 of urban alkaline-soluble organic matters (ASOM) collected from the
city of Aveiro, Portugal (Matos et al., 2015).
The maximum fluorescence intensity ($F_{max}$) was calculated by multiplying the
maximum excitation loading and maximum emission loading for each component by
its score (Murphy et al., 2013). Generally, changes in the relative abundance of a
component ($F_{max}/\sum F_{max}$) could indicate changes in its overall importance, which had
been successful applied to study the origins of chromophores (Yan and Kim,
2017;Chen et al., 2017a;Chen et al., 2016b;Wu et al., 2019). In this study, the relative
abundances of fluorescent components in different types of samples were highly
variable, depending on the sources (Figure 3a). P1 component accounted for an
average of $34 \pm 4.7\%$ of the total fluorescence intensities in the case of tunnel aerosols,
which was higher than BB aerosols (mean $\pm$ SD: $19 \pm 4.8\%$), CC aerosols ($14 \pm 3.8\%$)
and vehicle exhaust particles ($17 \pm 1.0\%$). The difference of P1 component between
tunnel aerosols and vehicle exhaust particles indicated P1 component had an aged
vehicle exhaust origin. In contrast, the fluorescence of P6 component was weak in all
the samples, but higher in vehicle emissions ($9.4 \pm 2.3\%$) than in BB and CC aerosols
(both 2.5%). P5 component was more intense in vehicle exhaust particles ($30 \pm 1.6\%$)
than in other sources. P2 component was abundant only in BB aerosols ($33 \pm 11\%$),
but not in vehicle emissions, which suggested that some structures responsible for this
chromophore could not exist in vehicle emissions. P4 component was the more
abundant chromophore in CC aerosols ($34\% \pm 7.7\%$) and vehicle emissions ($29 \pm
5.9\%$), especially in vehicle exhaust particles ($38 \pm 1.1\%$). In contrast, P4 component
in BB aerosols was weak ($11\% \pm 7.9\%$), indicating a fossil origin. P3 component was

almost equal across all samples. The possible reason was that P3 component was

similar to the peak of tryptophan-like compounds which were common in practically

all published models and were likely to be found in almost all sources (Yu et al.,

2015).

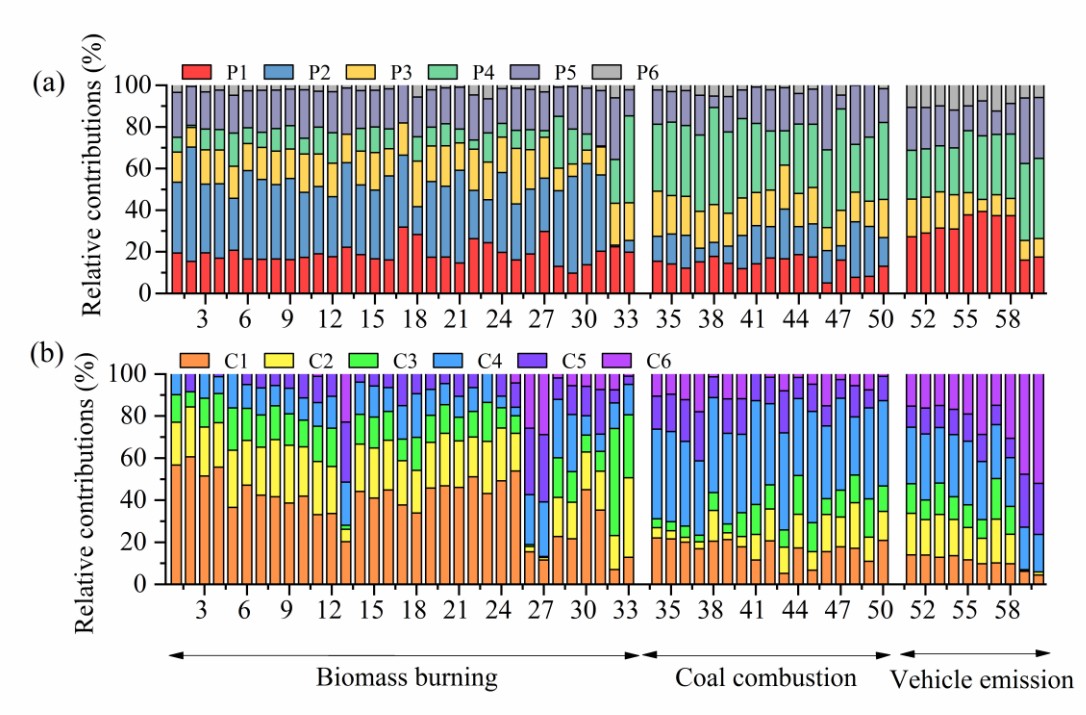

**Figure 3.** The relative contributions of each PARAFAC component of WSOC (a) and MSOC (b) from

the three sources.

The relative intensities of fluorescent components in MSOC exhibited similar

characteristics to WSOC (Figure 3b). C1 component was the substances with more

intense in the case of BB aerosols (38% ± 14 %) than the other sources. C2

component was enriched in BB aerosols (21% ± 6.9 %) and tunnel aerosols (17% ±

6.9 %) than those in CC aerosols and vehicle exhaust particles. In addition, C2

exhibited a difference between bituminous CC and anthracite combustion aerosols, as

well as tunnel aerosols and vehicle exhaust particles, indicating C2 component could

be used to identify these sources. C4 component was intense in CC aerosols (41 ±

6.0%) and vehicle emissions (26 ± 4.4%). C3 component was not abundant among the

three sources and not observed in the vehicle exhaust particles, suggesting not a fresh

vehicle-exhaust emission origin. C5 and C6 components were more intense in vehicle

exhaust particles (25 ± 6.8% and 50 ± 6.8%, respectively), suggesting they were the
primary vehicle emission chromophores. The last study observed that the relative
abundances of various chromophores in aerosols with different particle sizes were
different (Chen et al., 2019). Therefore, the fluorescence technique is sensitive for
chromophores with different sources, sizes, and chemical structures and so on. In
summary, the variation of the fluorescent components from different sources obtained
by EEM-PARAFAC method could be helpful to the source apportionment of BrC in
environmental applications.
**3.3 Molecular composition detected by FT-ICR MS**
The molecular compositions of WSOC and MSOC extracted from BB and CC
aerosols, and vehicle emissions were determined by negative ESI-FT-ICR MS. ESI is
a soft ionization method, and it can only ionize polar organic compounds, hydrophilic
molecules (Wozniak et al., 2008), but nonpolar or less polar compounds such as
polycyclic aromatic hydrocarbons (PAHs) and saturated hydrocarbons are not easily
ionized by ESI (Lin et al., 2018). In addition, ESI− cannot detect the N-heterocyclic
alkaloid compounds (Laskin et al., 2009). Thus, this study mainly discussed these
readily ionizable polar organic compounds by ESI−.
Figure 4 showed the reconstructed negative-ion ESI FT-ICR mass spectra of
WSOC for the six selected samples. Lots of peaks with an intensive mass ranges
between m/z 150 and 600 were showed in the mass spectra, with the most massive
numbers of ions within the ranges of m/z 200−400. Additionally, more formulas were
detected in BB aerosols (total 7708) than CC aerosols (5305) and vehicle emissions
(4047) (Table 2), suggesting a higher observed chemical complexity (i.e., the
observed peaks). According to the intensity of each ion, the average molecular
formulas of WSOC in the six aerosol samples were calculated and listed as follows:
$C_{18.7}H_{23.5}O_{6.99}N_{0.73}S_{0.09}$, $C_{19.9}H_{21.5}O_{7.65}N_{0.34}S_{0.03}$, $C_{16.1}H_{13.3}O_{5.37}N_{0.68}S_{0.23}$,
$C_{15.2}H_{13.7}O_{4.24}N_{0.45}S_{0.41}$, $C_{13.4}H_{18.0}O_{7.52}N_{0.45}S_{0.40}$, and $C_{17.3}H_{21.1}O_{5.65}N_{0.53}S_{0.08}$ for Musa,
Heave, anthracite, bituminous coal, tunnel, and vehicle exhaust, respectively. The BB
aerosols had higher contents of C and H, while the CC aerosols and tunnel aerosol had
higher contents of S.
**Table 2.** Number of formulas in each compound category and the average values of elemental
ratios, molecular weight (MW), double-bond equivalents (DBE), and aromaticity index ($AI_{mod}$) in
WSOC from the six aerosol samples.

| Samples | Elemental composition | Number of formulas | $MW_w$ | $DBE_w$ | $AI_{mod,w}$ | $O/C_w$ | $H/C_w$ | $DBE/C_w$ |
|---|---|---|---|---|---|---|---|---|
| Musa | Total | 4534 | 372.55 | 8.36 | 0.33 | 0.37 | 1.25 | 0.45 |
| | CHO | 1504 | 367.73 | 8.08 | 0.32 | 0.38 | 1.25 | 0.43 |
| | CHON | 2375 | 384.06 | 9.31 | 0.39 | 0.34 | 1.22 | 0.48 |
| | CHOS | 329 | 320.06 | 4.59 | 0.15 | 0.51 | 1.46 | 0.34 |
| | CHONS | 323 | 358.24 | 5.04 | 0.12 | 0.51 | 1.51 | 0.35 |
| Hevea | Total | 3174 | 387.05 | 10.32 | 0.42 | 0.38 | 1.08 | 0.52 |
| | CHO | 1610 | 377.86 | 10.06 | 0.42 | 0.38 | 1.08 | 0.51 |
| | CHON | 1408 | 409.40 | 11.29 | 0.46 | 0.39 | 1.05 | 0.55 |
| | CHOS | 108 | 376.68 | 7.00 | 0.23 | 0.38 | 1.32 | 0.39 |
| | CHONS | 48 | 410.33 | 5.08 | 0.09 | 0.47 | 1.60 | 0.30 |
| Anthracite | Total | 3930 | 308.65 | 10.82 | 0.65 | 0.33 | 0.83 | 0.67 |
| | CHO | 990 | 283.07 | 11.06 | 0.67 | 0.28 | 0.77 | 0.67 |
| | CHON | 1808 | 323.71 | 11.67 | 0.71 | 0.34 | 0.81 | 0.69 |
| | CHOS | 464 | 308.97 | 8.73 | 0.49 | 0.36 | 0.95 | 0.59 |
| | CHONS | 668 | 332.83 | 8.99 | 0.52 | 0.46 | 0.95 | 0.63 |
| Bituminous coal | Total | 1375 | 282.91 | 9.63 | 0.61 | 0.28 | 0.90 | 0.63 |
| | CHO | 399 | 259.21 | 10.40 | 0.66 | 0.22 | 0.82 | 0.65 |
| | CHON | 411 | 267.68 | 9.92 | 0.69 | 0.27 | 0.86 | 0.67 |
| | CHOS | 302 | 324.65 | 9.51 | 0.49 | 0.28 | 0.99 | 0.57 |
| | CHONS | 263 | 299.28 | 7.98 | 0.56 | 0.43 | 0.98 | 0.63 |
| Tunnel | Total | 2746 | 317.68 | 5.68 | 0.35 | 0.56 | 1.34 | 0.42 |
| | CHO | 803 | 298.29 | 7.69 | 0.49 | 0.50 | 1.06 | 0.54 |
| | CHON | 1049 | 340.18 | 7.50 | 0.38 | 0.51 | 1.22 | 0.49 |
| | CHOS | 508 | 310.74 | 2.73 | 0.03 | 0.59 | 1.71 | 0.23 |
| | CHONS | 386 | 337.90 | 2.78 | 0.46 | 0.81 | 1.77 | 0.25 |
| Vehicle exhaust | Total | 1301 | 327.71 | 7.96 | 0.41 | 0.33 | 1.22 | 0.46 |
| | CHO | 561 | 311.62 | 8.02 | 0.43 | 0.30 | 1.19 | 0.46 |
| | CHON | 673 | 320.62 | 7.28 | 0.41 | 0.40 | 1.27 | 0.47 |
| | CHOS | 63 | 467.88 | 11.88 | 0.36 | 0.19 | 1.19 | 0.44 |
| | CHONS | 4 | 438.78 | 2.21 | 0 | 0.46 | 1.97 | 0.12 |

In this study, these identified molecular formulas were classified into four main
compound groups based on their compositions: CHO, CHON, CHOS, and CHONS.
CHO compounds refer to the compounds that contain carbon, hydrogen, oxygen, and
the other compound groups that are defined analogously. The relative abundances of
the four compound groups were calculated by the magnitude of each peak divided by
the sum of magnitudes of all identified peaks and showed in Figure 4. CHO was the
most abundant component in WSOC, accounting for 43%−69% of total intensities of
BB aerosols, 36%−37% of CC aerosols, and 36%−47% of vehicle emissions,
respectively. CHO in BB and CC aerosols were lower than those of mass spectra from
simulated combustion experiments (BB (53%−72%) and CC (43%)) (Song et al.,
2018). Generally, CHO formulas were consistent with species reported previously as
lignin-pyrolysis products (Fleming et al., 2017), and they detected this fraction with
43.1% ± 14.6% in brushwood-*chulha* cook firers. CHON was abundant in the three
sources. This result was different from the findings that CHON species had a higher
percentage in BB smoke and were not abundant in CC smoke (Song et al., 2018). The
high fraction of CHON in CC aerosols could be due to that the N-containing
compounds in the BB smoke $PM_{2.5}$ come from the nitrogen content in the fuels
(Coggon et al., 2016), and the contents in coal fuels were comparable to biomass fuels
(See Table S1 and S2). However, S-containing compounds were more abundant in CC
aerosols (9.2%−21% for CHOS and 13%−20% for CHONS, respectively) and tunnel
aerosol (24% for CHOS and 16% for CHONS, respectively) than those in BB aerosols
(2.0%−5.6% for CHOS and 0.62%−3.7% for CHONS, respectively) and vehicle
exhaust particle (7.5% for CHOS and 0.25% for CHONS, respectively), consistent
with the previous studies (Song et al., 2018;Wang et al., 2017). ESI was more
efficient in ionizing S-containing compounds and most of them were selectively
ionized by ESI−, suggesting that they were polar species such as organosulfates (Lin
et al., 2018). Our study reported that S-containing compounds in WSOC were
associated with CC emissions by combining with carbon isotope data ($^{14}C$) (Mo et al.,
2018). Furthermore, the relative abundances of group species in CC aerosols and
tunnel aerosol were similar to those of water extracts in the hazy day (Jiang et al.,
2016), indicating both sources could be the important contributors of haze. However,
differences between tunnel aerosol and vehicle exhaust particle were observed,
indicating S-containing compounds in the tunnel aerosol were more secondary
formation.

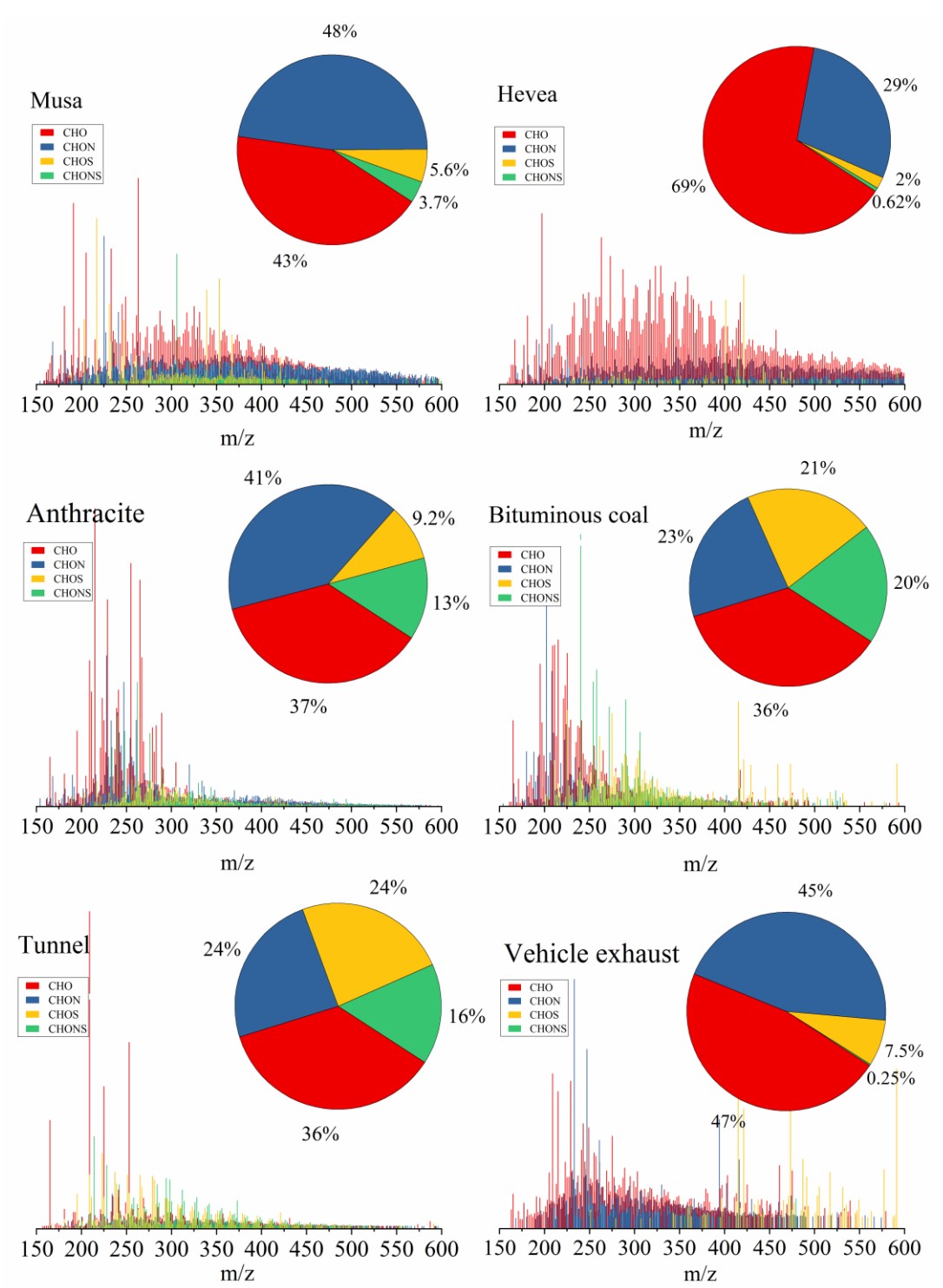

**Figure 4.** Negative ESI FT-ICR mass spectra of WSOC from the six aerosol samples. Different formula groups were color-coded. The six pie charts showed the relative intensities of different formula groups.

Van Krevelen (VK) diagram is a useful tool that provides a visual graphic display of compound distribution, and to some extent, use to qualitatively identify different composition domains in organic mixtures (Song et al., 2018;Lv et al., 2016;Smith et al., 2009). In this study, each source showed similar VK patterns. As

shown in Figure S9, Musa and Hevea burning had a VK diagram similar to that of WSOC in straw burning and fog water (Schmitt-Kopplin et al., 2010;Mazzoleni et al., 2010). S-containing compounds in tunnel aerosol with high O/C and H/C ration were similar to the aerosol-derived WSOC in New York and Virginia (Wozniak et al., 2008). Six dominate domains were identified in WSOC, including lignins, carbohydrates, tannins, proteins, condensed aromatic, and unsaturated hydrocarbons. In addition, results showed compounds observed in CC aerosols had lower H/C and O/C ratios than those in BB aerosols and vehicle emissions, indicating a higher unsaturated degree and lower oxidation level. There were compounds outside the specified regions, which had a high H/C ratio ($\geq2.2$), and DBE = 0 correspond to saturated oxygenated species and could be some long-chain polyalcohols (Lin et al., 2012a).

The mass spectra of MSOC exhibited differences from WSOC (Figure S10), especially in BB aerosols and vehicle emissions that exhibited larger m/z in the range of 350−600. The detected formulas in MSOC were much lower than those in WSOC, with the total number of 4502, 3628, and 1069 for BB, CC, and vehicle emission aerosols, respectively (Table S5). The reason could be due to that ESI− can efficiently ionize the polar compounds, and the methanol extracts after water-extracted may contain more moderate- and low-polarity compounds that were not easily ionized. The average molecular formulas were $C_{26.9}H_{46.2}O_{4.27}N_{0.24}S_{0.02}$, $C_{23.3}H_{34.9}O_{5.18}N_{0.20}S_{0.02}$, $C_{18.2}H_{19.2}O_{4.24}N_{0.92}S_{0.03}$, $C_{22.4}H_{20.7}O_{3.01}N_{0.38}S_{0.05}$, $C_{22.6}H_{44.1}O_{5.70}N_{0.74}S_{0.11}$, and $C_{25.2}H_{48.5}O_{4.86}N_{0.58}S_{0.08}$ of MSOC in the six aerosol samples, respectively, showing higher C and H contents than their corresponding formulas of WSOC but a decreasing trend in O contents.

CHO and CHON were the main components in MSOC, accounting for about 90% of the total intensities (CHO plus CHON). CHO was the most abundant category observed in BB aerosols (78%−80%). The elemental compositions observed in CC aerosols were different between bituminous coal and anthracite combustion. The abundance of CHON in anthracite combustion was higher (73%), while the CHO in bituminous combustion was higher (60%), which was consistent with the corresponding WSOC. It might be due to higher N content and lower O content of anthracite than that of bituminous coal (see Table S2). However, CHON in BB aerosols (18%−20%) exhibited lower abundances than those in CC aerosols and vehicle emissions. S-containing compounds were not abundant in MSOC. It may be

due to that the combination of S element and O atom show higher polarity.

Figure S11 showed the VK diagram of MSOC in the six aerosol samples. More formulas in BB aerosols exhibited two distinct groups with H/C of 1.4−2.2 and 0.6−1.4 vs. O/C of 0.1−0.5, in three domains (lignins, proteins, and lipids). Compounds in CC aerosols with lower H/C and O/C ratios were dominant in the domains of lignins and condensed aromatic, especially in bituminous CC aerosol with more unsaturated hydrocarbon. Tunnel aerosol showed a wide range of O/C in S-containing compounds and a wide range of H/C in non-S-containing compounds. In contrast, compounds in vehicle exhaust particle had a wide range of H/C but a narrow O/C ratio. The VK diagram with fewer S-containing compounds in vehicle exhaust particle showed a similar characteristic to the distribution of non-S-containing compounds in tunnel aerosol, indicating the difference was mainly due to the S-containing compounds.

Table 2 and S5 presented the relative abundance weighted molecular weight ($MW_w$), double bonds equivalence ($DBE_w$), and modified aromaticity index ($AI_{mod,w}$) of WSOC and MSOC, respectively. DBE was used as a measure of unsaturated level in a molecule, and $AI_{mod}$ could be used to estimate the fraction of aromatic and condensed aromatic structures (Song et al., 2018;Lv et al., 2016;Koch and Dittmar, 2006). BB aerosols had higher $MW_w$ values than CC and vehicle emissions in WSOC. Further, higher $DBE_w$ and $AI_{mod,w}$ values were observed in CC aerosols than the others. MSOC had higher $MW_w$ but lower $AI_{mod}$ values than the corresponding WSOC. Furthermore, CHO and CHON compounds had higher $DBE_w$ and $AI_{mod,w}$ values than S-containing substances, consistent with the earlier results (Lin et al., 2012b;Lin et al., 2012a).

Figure S12 showed the fraction of $AI_{mod}$ values of WSOC in the six aerosol samples, where the formulas were classified according to their $AI_{mod}$ (aliphatic ($AI_{mod}$ = 0), olefinic ($0 < AI_{mod} \leq 0.5$) and aromatic ($AI_{mod} > 0.5$)). The results illustrated that the fraction of aromatic structure in non-S-containing compounds was higher than those in S-containing compounds. CC aerosols had a higher aromatic fraction than BB aerosols and vehicle emissions, especially in CHO and CHON (up to 89% of total ion intensities). In BB aerosols, the non-S-containing compounds had a high fraction of olefinic structure, following by aromatic structure, but the S-containing compounds had a higher aliphatic and olefinic structure than aromatic structure. A higher fraction of aliphatic in vehicle emissions was observed in the S-containing compounds

(especially in tunnel aerosol (exceed 81%)). These aliphatic S-containing compounds
might form by the precursors (long-chain alkanes) from vehicle emissions (Tao et al.,
2014), which had higher H/C and lower DBE values (see Table 2). However, the
previous study showed that AI must be regarded as the most conservation approach
and may result in an underestimate of the aromatic structures (Koch and Dittmar,
2006), which was observed in Beijing aerosols (Mo et al., 2018). Although $AI_{mod}$
identified more compounds as aromatic and condensed aromatic components than AI,
the $AI_{mod}$ may introduce uncertainties for individual molecules, which was
demonstrated by Koch and co-author.
For MSOC, the aromatic structure fractions in non-S-containing compounds
were higher than those in S-containing compounds, and the aromatic structure
fractions in CC aerosols were higher than those in BB aerosols and vehicle emissions
(Figure S13), which was consistent with WSOC. Furthermore, we found that the
fraction of aliphatic in MSOC was higher than that in WSOC, indicating more fat-like
compounds.
*Different chemical characteristics of BB, CC, and vehicle emissions*
Figure S14 plotted the Venn diagram of formulas in WSOC in the six aerosol samples
for determining the unique elements in the mass spectra. The previous study identified
the unique elements of water-soluble HULIS in simulated BB and CC smokes, which
presented different molecular characteristics between biomasses, as well as between
biomass and coal (Song et al., 2018). In this study, we combined more formulas of
different sources to determine the unique molecules and more limitations were set,
which would provide more identified characteristics for each source. 212 molecular
formulas were detected simultaneously in the six aerosol samples. It is noting that
without any further information, it is not possible to decide whether these common
formulas represent the same compounds. There were 112 of CHO unique molecules in
212 and 98 of CHON but only 2 of CHOS molecules. CHO compounds were
relatively small aromatic compounds with 8−20 C atoms and 3−8 O atoms and DBE
5−13 and multiple acidic polar functional groups (Figure S15). It is noting that lines
in Figure S15 indicate DBE reference values of linear conjugated polyenes $C_xH_{x+2}$
with DBE=0.5×C, and fullerene-like hydrocarbons with DBE=0.9×C, where the data
points inside this region are potential BrC chromophores (Lin et al., 2018). For
example, organic acids ($C_8H_6O_5$ (DBE=6)) was detected in urban $PM_{2.5}$ (Yassine et al.,
2012), as well as $C_9H_8O_5$ (6), $C_{14}H_{14}O_4$ (8), $C_{13}H_{14}O_5$ (7), which allowed them to
ionization in the ESI− mode and were identified as potential BrC chromophores. In
total, all of CHON compounds had O/N>2 (5.3±1.28, 2.5−8) (Figure S15), allowing
for the assignment of at least one nitro (-NO$_2$) or nitrooxy (-ONO$_2$) group and other
oxygen-containing groups (i.e., -OH and -COOH). Except for $C_{19}H_{41}O_7N$ (DBE=0),
the remaining compounds with DBE≥5 were suggested as nitro-aromatic and
nitrophenol derivatives (Mo et al., 2018;Lin et al., 2018). CHOS species only had two
formulas including $C_{18}H_{38}O_7S$ (0) and $C_{20}H_{38}O_7S$ (2). It was reported that O$_7$S groups
were the most abundant species class in CHOS identified in water extracts of PM$_{2.5}$
(Jiang et al., 2016).

There were more observed unique peaks of WSOC in BB aerosols (total 1947)

compared to CC aerosols (1583) and vehicle emissions (813). However, only 143 and
83 molecular were identified in bituminous CC and vehicle exhaust particle,
respectively. Among the observed compounds, 1353 and 1440 unique molecular
formulas were detected in combustion of Musa and anthracite, respectively, implying
a significant difference from the others. Figure 5 (a) showed the VK diagram of these
unique formulas of WSOC for each sample, where four regions were circled for
representing different sources. The results indicated that these unique compounds in
different sources had a distinctive chemical characteristic. That may be the reason that
resulted in variable fluorescent spectra in different sources (discussed above).
Additionally, the diagram showed that the unique molecules in CC aerosols were
located in the region with lower H/C and O/C, and vehicle emissions containing
tunnel aerosol and vehicle exhaust particle were located in two distinct regions.

Figure 6 showed plots of the DBE vs. the number of carbon atoms in the unique

molecular formulas of all aerosol samples. These compounds observed in BB aerosols
were largely CHO and CHON (CHO and CHON, 88%−93%) with C numbers ranging
from 6 to 40 and DBE ranging from 0 to 31, with no regular distribution. S-containing
compounds were the important components in the unique molecular formulas of CC
aerosols (CHOS and CHONS, 38%−75%) and vehicle emissions (CHOS and CHONS,
41%−66%). However, only 7%−12% of the total unique molecular formulas were
observed in BB aerosols. As shown in Figure 6, the region marked by blue box
denoted the high intensities of compounds in unique formulas of each sample. The
high-intensity compounds detected in Musa burning aerosol were mainly C number

from 14 to 24, DBE from 7 to 13, and two N atoms, such as $C_{20}H_{26}O_7N_2$ (9), $C_{18}H_{24}O_5N_2$ (8), $C_{22}H_{28}O_6N_2$ (10), $C_{19}H_{26}O_7N_2$ (8), $C_{21}H_{28}O_6N_2$ (9), $C_{14}H_{18}O_3N_2$ (7), $C_{24}H_{30}O_8N_2$ (11), and $C_{21}H_{24}O_5N_2$ (11) and so on. Instead of Musa, the abundant compounds in Hevea burning were mainly $C_{24}H_{22}O_9$ (14), $C_{28}H_{28}O_{11}$ (15), and $C_{28}H_{26}O_{11}$ (16), and so on. Although the difference between burning of Musa and Hevea appeared, the VK diagram (Figure 5) did not show distinct changes. The high-intensity compounds in anthracite combustion with lower C atoms than in bituminous CC, were main $C_{14}H_8O_5N_2$ (12), $C_{12}H_{11}O_4NS$ (8), $C_{12}H_{10}O_8N_2$ (9), while in bituminous CC were main $C_{28}H_{28}O_4S$ (15) and its homolog of $C_{27}H_{26}O_4S$ (15), and $C_{19}H_{16}O_3S$ (12). The abundant compounds in tunnel aerosol had a lower unsaturation degree, such as $C_4H_9O_7NS$ (1), $C_5H_{11}O_7NS$ (1), $C_7H_{14}O_5S$ (1). In vehicle exhaust particle, the high intensity of compounds was one fraction with low C atoms and DBE ($C_{21}H_{40}O_8N_2S$ (3), $C_{26}H_{46}O_3S$ (4)), and the other fraction with high C atoms and DBE ($C_{32}H_{34}O_8S$ (16), $C_{30}H_{34}O_5S$ (14)). These findings are essential because these unique molecular formulas in different sources may have specific chemical composition, which would help the source apportionment of aerosols.

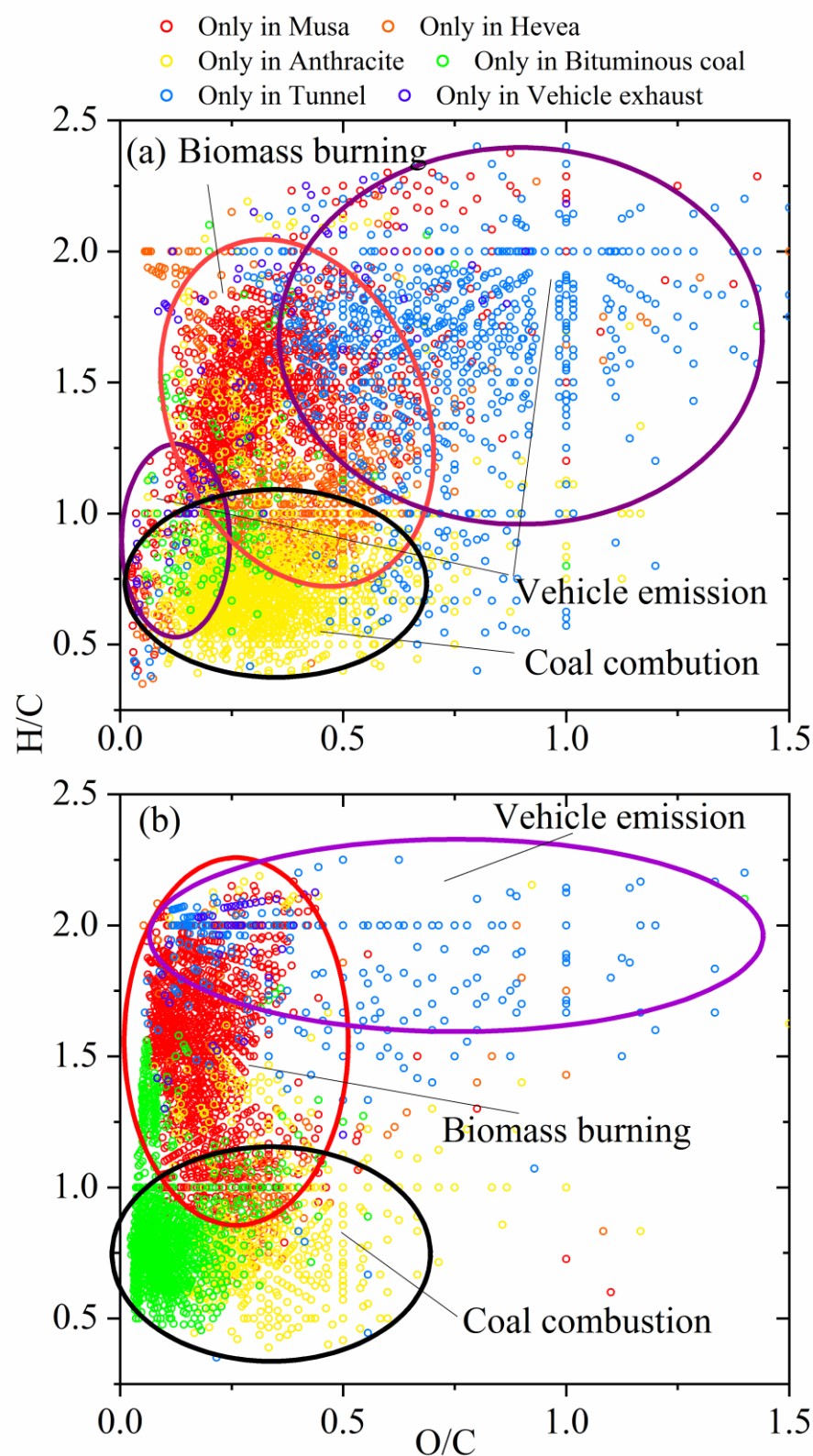


**Figure 5.** A Van Krevelen diagram of WSOC (a) and MSOC (b) from the six aerosol samples.

Different color indicates unique formulas detected in each sample.

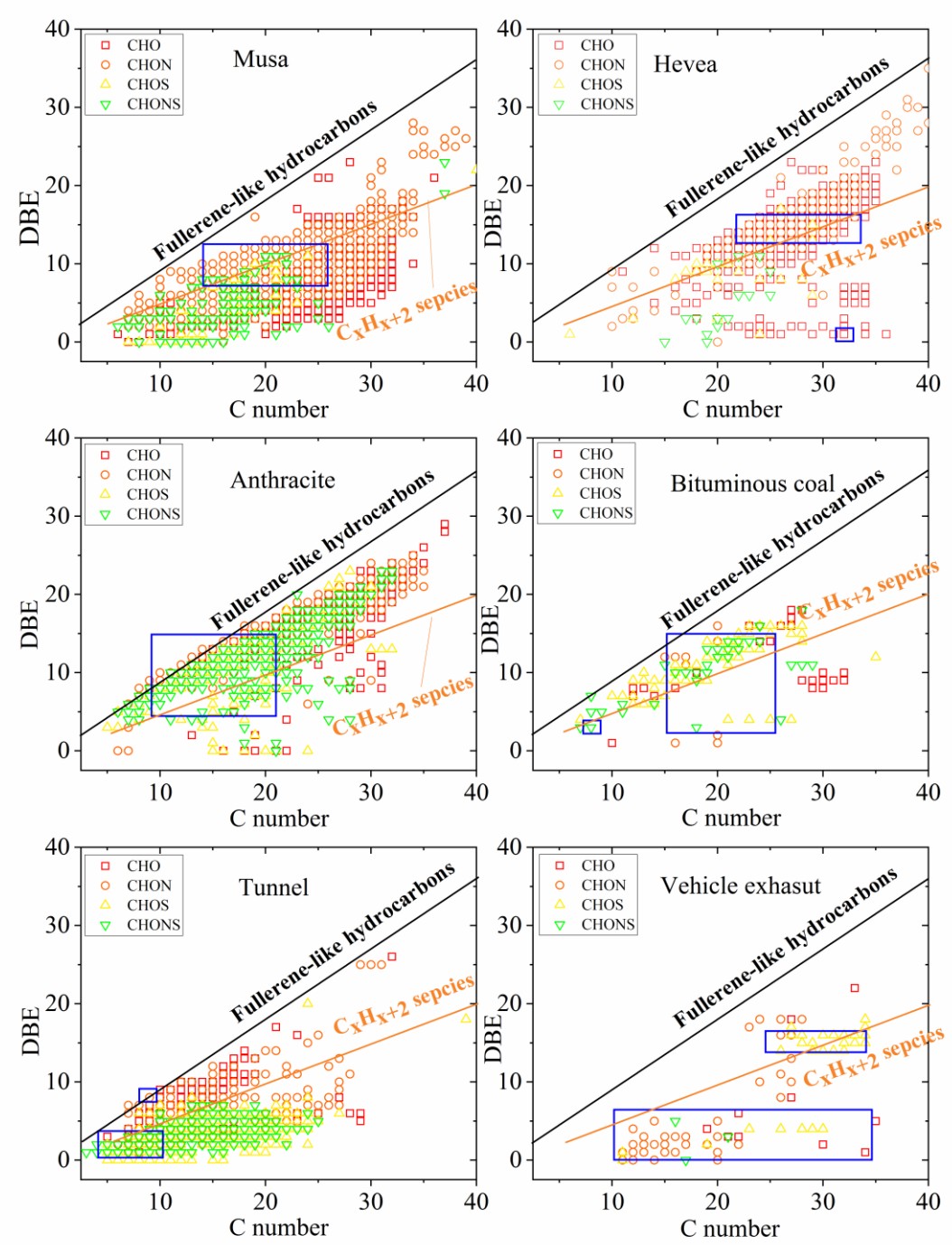

**Figure 6.** DBE vs. C number for unique molecular compounds of WSOC from the six aerosol
samples. Lines indicate DBE reference values of linear conjugated polyenes $C_xH_{x+2}$ with
DBE=0.5×C, and fullerene-like hydrocarbons with DBE=0.9×C. The regions marked by blue box
denoted the high intensities of compounds.
Compared to WSOC, Figure S16 showed fewer compounds in common in
MSOC for the six aerosol samples. There were only 44 compounds common to the six
aerosol samples. A total of 26 and 14 of the 44 formulas were CHO and CHON,
respectively, but only 4 of the 44 formulas were S-containing compounds. As shown

in Figure S17, there were only three compounds ($C_{17}H_{10}O_2$ (13), $C_{18}H_{14}O$ (12), $C_{18}H_{12}O_2$ (13)) in CHO group, and one compound ($C_{14}H_{11}O_4N$ (10)) in CHON group inside the potential BrC region. The remaining compounds had a high C number (18−35), low O atoms (1−7), and low DBE (0−2), suggesting that they mostly had fatty acid structures.

These unique molecules in VK diagram also showed similar results comparing to WSOC (Figure 5 (b)), further confirming the special characters in different sources. Expect for tunnel aerosol (about 50%), these unique formulas in the BB aerosols, CC aerosols, and vehicle exhaust particle was dominant by CHO- and CHON-groups (Figure S18). The high-intensity compounds were $C_{35}H_{69}O_5N$ (2), $C_{38}H_{76}O_4$ (1) for Musa burning; $C_{26}H_{22}O_7$ (16), $C_{28}H_{26}O_7$ (16) for Hevea burning; $C_{14}H_{12}O_6N_2$ (10), $C_{17}H_{14}O_5N_2$ (12) for anthracite combustion; $C_{23}H_{16}O$ (16), $C_{24}H_{18}O$ (16), $C_{24}H_{14}O$ (18) for bituminous CC; $C_4H_9O_7NS$(1), $C_{24}H_{42}O_3S$ (4), $C_8H_{16}O_5S$ (1) for tunnel aerosol, and $C_{26}H_{37}O_5NS$(7), $C_{22}H_{46}O_7$ (0) for vehicle exhaust particle, respectively.

**3.4 Link of molecular compositions and optical properties**

In the above statements, we discussed the light absorption and fluorescence properties from aerosols in the three different sources. The light absorption capacity of WSOC and MSOC was essential to assess the evolution of BrC, and fluorescence spectra were sensitive to different sources and could help for the source apportionment of BrC. In addition, we evaluated the molecular compositions of the three sources. Therefore, understanding the factors affecting the optical properties of BrC is important. It was reported that the MAE in the BB experiments depended largely on burning conditions (Chen and Bond., 2010) and in the CC experiments depended on coal maturity (Li et al., 2018). Chen et al. (2017b) illustrated that the higher light absorption capacity was associated with the low- and medium-polarity fractions that contained aromatic and polar functional groups (O or both O and N atoms). Sources play an important role in light absorption capacity, consistent with our current study. The $MAE_{365}$ values of WSOC in highly BB-impacted areas were two times higher than in low BB-impacted areas in the Southeastern United States (Hecobian et al., 2010). Atmospheric aging has a significant effect on the light absorption capacity of BrC (Li et al., 2019), but the mechanism involved is very complex. The response of the light absorption capacity of different types of BrC to aging is highly variable, and enhancement or reduction in the light absorption capacity

of BrC is possible (Li et al., 2019). These results indicated that light absorption
capacity might be affected by various factors. In this study, the higher $MAE_{365}$ values
were observed in BB and CC aerosols than vehicle emissions, and the chemical
structures and unsaturation degree of different sources were discussed. Next, we
further discussed the relationship between optical properties and chemical structures
below.
Before discussing their relationship, we firstly determined these compounds that
were potential to absorb light radiation based on the above statement to reduce the
influence of non-absorbing substances (Lin et al., 2018). Mo et al. (2018) reported
that $MAE_{365}$ of HULIS in aerosols was affected by oxidation level and unsaturation
degree. In this study, the $MAE_{365}$ had no significant correlation with O/C, indicating
that light absorption capacity does not appear to be affected by their oxidized
properties in the source emission aerosols. Instead of O/C, the $MAE_{365}$ had a
significant positive correlation with the average DBE and MW, respectively (Figure
7), suggesting the unsaturation level and MW played a vital role in the light
absorption capacity of source samples. Field experiments indicated that the majority
of absorption was the larger molecules (>500 Da) (Di Lorenzo et al., 2017). It is
crucial to knowledge the relationship between light absorption of source samples and
their molecular compositions due to the compounds in fresh emissions that may
undergo a secondary process and introduce more uncertainty for their optical
properties.

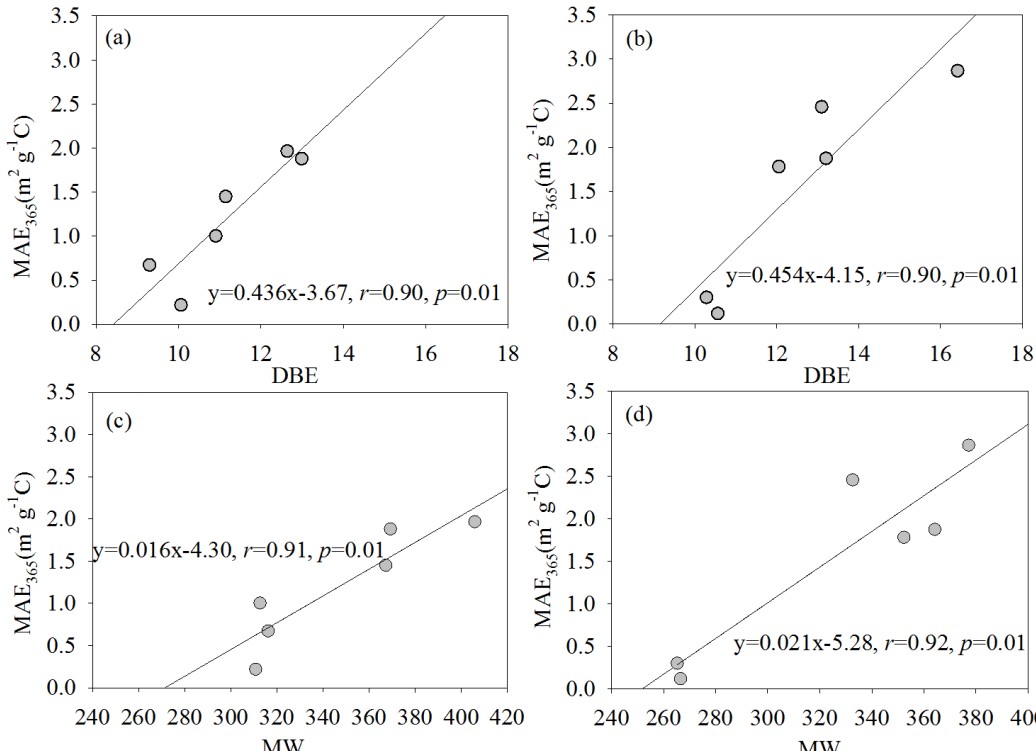

**Figure 7.** Relationships between DBE and MW of the potential BrC molecules and the $MAE_{365}$ of
WSOC (a, c) and MSOC (b, d) from the six aerosol samples, respectively.

Fluorescence spectra can provide more information than UV-vis spectra. A red
shift in the excitation/emission maximum could indicate increased aromaticity and
higher molecular weight (Ghidotti et al., 2017). Field observation had demonstrated
that chromophore components were associated with chemical structures (Chen et al.,
2016b;Chen et al., 2016a;Stubbins et al., 2014). Chen et al. (2016b) illustrated that the
fluorescent components of HULIS-1 and HULIS-2 were correlated positively with
$CO^+$ and $CO_2^+$, and $C_xH_y^+$ and $C_xH_yO_1^+$ groups ions, respectively, using the
correlation analysis of the relative intensities of ion groups in the high-resolution
aerosol mass spectrometers (HR-AMSs) and relative contents of fluorescence
components. In another study, Chen et al. (2016a) demonstrated that fluorescent
components had strong links with chemical groups in the Fourier transform infrared
(FT-IR) spectra, including the oxygenated functional groups (nonacidic carbonyl C=O
and carboxylic COOH groups), aliphatic C-H group, amine C-NH2, and alcohol C-OH
groups. The chromophores are sensitive to sources, and it is very important to
understand the molecular composition of chromophores for classification and source
apportionment of atmospheric BrC. However, the ESI− cannot ionize the most typical
BrC chromophores such as O-heterocyclic PAHs (O-PAHs), N-heterocyclic PAHs

(N-PAHs) (Lin et al., 2018), which was not enough to discuss the relationship between the fluorescence spectra and molecular composition. The combination of atmospheric pressure photoionization (APPI+ and APPI−) and ESI (+ and −) may provide more ionizable compounds, but these techniques were not with the scope of our study.

**4 Conclusions**

We conducted comprehensive measurements on light absorption, fluorescence, and molecular compositions of dissolved BrC derived from smoke particles during the simulated combustion of biomass and coal, as well as vehicle emission aerosols. We observed BB and CC aerosols had high $MAE_{365}$ values than vehicle emissions, on average, $1.6 \pm 0.55$, $1.3 \pm 0.34$, $2.0 \pm 0.75$, and $0.71 \pm 0.30$ $m^2$ $g^{-1}$ C for BB, anthracite combustion, bituminous CC and vehicle emission aerosols, respectively. In addition, BrC emitted from BB ($2.3 \pm 1.1$ $m^2$ $g^{-1}$ C) and bituminous CC ($3.2 \pm 1.1$ $m^2$ $g^{-1}$ C) in MSOC exhibited stronger light absorption capacity than those in WSOC, but opposite results were found in anthracite combustion aerosols ($0.88 \pm 0.74$ $m^2$ $g^{-1}$ C) and vehicle emissions ($0.26 \pm 0.09$ $m^2$ $g^{-1}$ C). EEM combining with PARAFAC analysis determined six types of fluorescent components that were assigned as two HULIS-1 (P1, and P6), three PLOM (P2, P3, and P5), and one undefined (P4) in WSOC from the three sources. The relative intensities of the fluorescent components mainly depended on the different types of sources. For example, HULIS-1 was abundant in tunnel aerosols, P2 was more intense in BB aerosols but not observed in vehicle emissions, P4 was intense in CC aerosols and vehicle emissions, P5 was more abundant in fresh vehicle exhaust particles; although P3 was not abundant it was ubiquitous in all tested aerosols. Similar to WSOC, six fluorescent components were identified in MSOC. Although the methanol-soluble chromophores were poorly understood, different characteristics were observed in different sources.

FT-ICR mass spectra showed that the m/z of the mainly compounds with m/z 200−400 in WSOC and MSOC was m/z 350−600 (except for CC aerosols), respectively. CHO and CHON were the main components in the six aerosol samples, but S-containing compounds were more abundant in CC and tunnel aerosols than BB aerosols and vehicle exhaust particles in WSOC. Similarily, MSOC mainly contained CHO and CHON species but fewer S-containing compounds. BB aerosols had higher CHO species in MSOC but showed lower CHON than CC aerosols and vehicle

emissions. Ven diagram showed that CC aerosols had more unsaturation degree and low oxidation level than the other two sources. This finding was further confirmed by a higher fraction of aromatic in CC aerosols. Unique formulas determined by Venn diagram showed certain specific chemical characteristics in VK diagram. BB aerosols emitted unique formulas with more CHO and CHON (88%−93%), while CC aerosols and vehicle emissions contained more S-containing compounds (38%−75% and 41%−46%, respectively). The relationship between optical properties and chemical structures showed the light absorption capacity was positively associated with an unsaturation degree and MW in the source emission samples. Our study illustrated the important roles of sources in light-absorbing BrC and molecular compositions, and the EEMs-based and molecular-characteristic-based method for classification and source apportionment of chromophores in atmospheric aerosols.

*Data availability.* The data used in this study are available upon request; please contact Gan Zhang (Zhanggan@gig.ac.cn) and Jun Li (junli@gig.ac.cn)

*Supplement.* The supplement related to this article is available.

*Author contributions.* JT, GZ, JL, and YC designed the experiment. JT and MC carried out the measurements and analyzed the data. JT, TS, YH, and HJ organized and performed the samplings. JT (Jianhui Tang) and BJ supported the fluorescence and FT-ICR MS instrument. JT wrote the paper. JL, YM, JS, PP, and GZ reviewed and commented on the paper.

*Competing interests.* The authors declare that they have no conflict of interest.

*Acknowledgements.* This study was supported by the Natural Science Foundation of China (NSFC; Nos. 41430645 and 41773120), the National Key R&D Program of China (2017YFC0212000), the International Partnership Program of Chinese Academy of Sciences (Grant No.132744KYSB20170002), and Guangdong Foundation for Program of Science and Technology Research (Grant No. 2017B030314057).

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
