# Peer review of "Molecular compositions and optical properties of dissolved brown carbon in"

_Atmospheric Chemistry and Physics, 2019_

## Referee Comment (RC1) · Anonymous Referee #1 · 21 Aug 2019

This manuscript describes measurements of the fluorescence of atmospheric WSOC, classifying it into separate types using parallel factor analysis, and attempting to correlate these types with high-resolution mass spectrometry data. The measurements of a very good technical quality and will be of interest to those who study aerosol fluorescence, organosulfates, or organonitrates. The work will also support future source

apportionment of aerosol by fluorescence measurements. However, some interpretations, conclusions and assertions are not adequately supported by the data presented. For this reason, major revision of the discussion of the mass spectrometry results (and the second half of the abstract) is needed. The work is potentially publishable after addressing the comments below.

The citation of previous work in the manuscript has some gaps. For example, it is odd that the manuscript comments on similarities between HULIS and terrestrial humic acids (line 70) without citing the authoritative review on this subject by Graber and Rudich.(1) One A. Laskin paper in the Results section is erroneously cited as "Alexander et al. 2009".

The authors fundamentally assume that a correlation between a fluorescent component and a set of MS formulas (like "CHON") means that they are determining the molecular compositions of the fluorescent molecules. This may not be true. Molecules with high electrospray ionization efficiencies are not necessarily the same as those with high absorbance or fluorescence. It would therefore be fortuitous if major ESI ions were the same ones responsible for observed absorbance or fluorescence, without extensive and identical chromatographic separation before each technique.(2, 3) Furthermore, it has been shown that many non-polar brown carbon components in biomass burning aerosol cannot be ionized by ESI.(3) The manuscript should discuss these issues.

The interpretation of functional groups from the ESI-MS data should be explained more clearly (line 376). The authors try four different methods (Tables S9 – S16), but it is difficult to understand the differences between them.

The authors should explicitly describe and justify their assumptions in assigning functional groups to formulas. For example, it appears that all compounds with S:O ratios of 1:4 have been assigned as organosulfates, while higher ratios are assigned as organosulfates with additional oxygen functional groups, and lower ratios are assigned to sulfonates. The following chemical arguments suggest that these assignments are

not only arbitrary, but incorrect. Sulfonates form from S(IV) + carbonyl reactions, and these reactions also generate products with S:O ratios of 1:4, but with a C-S bond. Thus, this reviewer would argue that the authors' use of S:O ratios to distinguish between sulfonates and organosulfates is invalid. Furthermore, organosulfate production is thought to require acid catalysis at very low pH. The measured near-neutral pH of the WSOC extracts in this work suggests that acids have been mostly neutralized, and therefore organosulfate formation (via acid catalysis) appears to be unlikely. As another example, the assignment of C17H16O4 and C18H16O4 ions from the C2 group to "esters" could use further justification.

Line 63: The phrase "little structural information is available" is misleading. There have been several studies, most involving Alex Laskin, that determined detailed chemical structures in biomass burning aerosol extracts. Some of these studies are eventually cited in this manuscript, but they should be briefly summarized here.

Line 103: It is inappropriate to follow the statement "Concerns about the environmental and health effects of
vehicle emissions have existed for decades" with only a single citation from 2015, unless the citation is a comprehensive review. This under-referencing happens at several points in the introduction.

Line 278: It is stated that measured MAEs are higher in this study than in previous lab studies of biomass and coal burning aerosol. Some explanation for this difference should be given. There is no other discussion of MAEs in the manuscript.

Line 320: The meaning of this sentence is unclear. What "other" fluorescent components are referred to?

Lines 33, 413, 464, 514: The authors appear to treat negative and positive correlations the same way in their interpretations. If P1 and P6 are negatively correlated with H-CHOS, this would mean that P1 and P6 fluorophores are formed whenever H-CHOS is not present, but the authors go on to attribute P1 and P6 to H-CHOS here, in the conclusion, and in the abstract.

Line 35, 519-520: These conclusions are questionable. See earlier comments about esters and sulfonates versus organosulfates.

Tables S5 and S6: It seems inappropriate to report either averages or total intensities across different types of samples like this, without some justification.

Technical corrections

There are a fair number of grammatical errors in the manuscript, which are not listed here. Fortunately, the meaning usually remains clear.

Line 49: the authors refer to the "near UV and UV/visible ranges," which are overlapping. Do they mean "near UV and visible ranges"?

Line 162: this statement would be clearer if the difference were explained. Is it true that MSOC is different in this study because WSOC has already been removed?

Line 213: The authors should briefly discuss what kinds of compounds will be missed by negative mode ESI. Will N-heterocycles be detected?

Line 241: "componet" should be "component"

Line 366: What kinds of differences? Differences in fluorescence? The fluorescence of sample 36 is not shown in Figures S9 or S10.

Line 371: This description of axes appears to be describing Figures 3 and 4, but these figures are not mentioned until later.

Line 375: The wrong figure is referenced here, I think.

Figures 4 and 5 would be improved by labeling the color code as number of oxygen atoms on the graph, not just in the caption.

Lin 516: While no MS structural class was correlated with P2 and P3, the literature designation of these fluorescent peaks is certainly consistent with the prevalence of these peaks in biomass burning samples in this study. This support can be mentioned

here.

Figures S2 and S5: "resident" should be "residual"

Figures S9 and S10: It would save readers a lot of time searching around if these samples where labeled with their sources on this graph (e.g. "bituminous coal")

Figure S11 caption: Different regions are identified, but not labeled on the graph. It is not clear what the reader can do with this information without further graphing.

References Cited: 1. Graber, E. R.; Rudich, Y., Atmospheric HULIS: How humic-like are they? A comprehensive and critical review. Atmos. Chem. Phys. 2006, 6, 729-753. 2. Lin, P.; Aiona, P. K.; Li, Y.; Shiraiwa, M.; Laskin, J.; Nizkorodov, S. A.; Laskin, A., Molecular Characterization of Brown Carbon in Biomass Burning Aerosol Particles. Environ Sci Technol 2016, 50, (21), 11815-11824. 3. Lin, P.; Fleming, L. T.; Nizkorodov, S. A.; Laskin, J.; Laskin, A., Comprehensive Molecular Characterization of Atmospheric Brown Carbon by High Resolution Mass Spectrometry with Electrospray and Atmospheric Pressure Photoionization. Anal. Chem. 2018, 90, (21), 12493-12502.
* * *

---

## Referee Comment (RC2) · Anonymous Referee #2 · 31 Aug 2019

This manuscript by Tang et al. describes a detailed chemical analysis on atmospheric brown carbon (BrC) extracted from smoke particles samples. Particle samples were collected from biomass burning, coal combustion, and vehicular emissions. Filter samples were extracted by either water or methanol and were analyzed with emission excitation matrix (EEM) and FTICR-MS with ESI(-) ionization. Six components were

extracted from the EEM data using a parallel factor analysis method. A significant amount of effort was present to make correlations between these EEM components with functional groups determined with FTICR-MS. The authors concluded that correlations were observed between EEM components and certain functional groups, indicating that this method can be useful in source apportionment of BrC.

The topic of the manuscript is in-line with the scope of ACP, in particular, the importance of BrC in the atmosphere is emergent, but there is extremely limited chemical information on important individual chromophores. The manuscript is attempting to address this important question. However, I do not recommend publication in ACP in the current form. In addition to a few major scientific questions, I have significant concerns regarding the literary presentation of the manuscript. It requires a substantial refinement before it can be published in any journal. In particular, I found the manuscript very difficult to read due to ill-structured order of discussion, missing or repeated explanations for abbreviations, frequent references to the SI, as well as numerous grammatical and typological errors.

Major comments:

EEM and ESI(-) are powerful analytical methods, but I'm afraid that they are not quantitative enough to make meaningful correlation analysis. Light absorptivity should be the primary concern for BrC chromophores, but fluorescence intensity, which is the core of the analysis here, depends on a number of other factors. Meanwhile, ESI(-) is particularly sensitive to compounds with acidic hydrogens, but not to PAHs and other compounds unless they have a carboxylic group. I'm afraid that the positive correlation could be driven by the detection sensitivities of the two methods.

The authors use the FTICR-MS to rule out functional groups. Although the authors present a thoughtful interpretation of the FTICR-MS data, caution is required, as what MS provides is the elemental composition, not functional group information. For example, the chemical structures shown in Figure 5 do not contain any acidic functional

group, and I double if they can be detected by ESI(-).

Table 2 is a critical part of the manuscript, presenting the functional group assignment based on FTICR-MS data. However, no explanation is provided for the table at all in the manuscript. Is the left side of the table linked to the right side of the table? The categories shown on the right side of Table 2 (Lipids, proteins, etc.) seem very irrelevant to atmospheric particles, but no explanation is provided in the main text.

The authors have presented a huge amount of work in interpreting the EEM components, FTICR-MS data, as well as the correlation analysis. The authors deserve a lot of credit for doing such a full-bodied analysis. However, the current conclusions in the manuscript do not appear very helpful for the atmospheric chemistry community, other than demonstrating the heterogeneity and complexity of the system. The authors should reconstruct the discussion and conclusion with more atmospheric implications.

Minor Comments

The manuscript is titled as "BrC in smoke particles". I personally felt odd that vehicular emissions are also included as smoke particles.

I am not a specialist in PARAFAC and found it difficult to see the concepts and purpose of PARAFAC until the end of Section 2.5. I recommend the authors added an introductory statement for PARAFRAC either in Introduction of Section 2.5. For example: "The purpose of PARAFRAC is to extract X components from the EEM data based on . . ."

Regarding water vs methanol extraction. The objectives of investigating WSOC and MSOC is unclear. Is the purpose to investigate BrC with distinct polarities? Is it to investigate "fat-soluble" fraction (Line 271)?

Related to the previous point, a discussion is needed on why the WSOC and MSOC are so distinct. To my understanding, these two solvents should extract different, but somewhat overlapping classes of organic compounds.

Line 266 - It is a little confusing because Figure 2b is introduced before Figure 1 and

the PARAFRAC components. Can the authors consider making Figure 2b an individual figure? Also, to make an argument on MAE is 'higher' or 'lower', more statistics are needed. Instead of presenting Figure 2b as is, I would recommend using a more statistical approach, such as a box and whisker plot.

Line 292 - What is 'region IV'?

Paragraph starting Line 369. It is very confusing that the paragraph started with an introduction to DBE, but the topic rapidly changed to O/C and H/C. The authors should consider reordering the discussion here.

What is AImod?

Figure 1 - no color scale explanation. Is each graph normalized to its highest intensity? The readers cannot see the relative importance of the 6 components (i.e., are one or two components much more intense than others?)

Figures 4 and 5- the authors introduced a region between slope 0.5 and 0.9 on the DBC vs C plot (Line 370). Why not show these lines in Figure 4 and 5?

Technical Comments - there are more grammatical errors than listed here, please check.

Line 39 - the abbreviation of EEM is already introduced in Line 23.

Line 82 and Line 85 - Chen et al / Lee et al are repeated.

Line 110 - the abbreviation of EEM is already introduced in Line 64.

Line 138 -'difficult' to 'difficulty'

Line 141 - 'Every coal about 1 kg fuels was burned three times'. To 'Coal ($\sim$ 1 kg each) was burned in triplicate."

Line 144 - 'Additional' to 'Additionally,'

Line 151 - 'truck' to ' a truck' or 'trucks'

Line 162 - 'MSOC fraction from the methanol extract' is redundant.

Line 164 - 'um'

Line 237 - 'Additional' to 'Additionally,'

Line 274 to 277 - MAE was higher in methanol extract for biomass burning and coal samples. I could not follow why that indicates a greater variation in chemical composition in MSOC.

Line 352 - 'abundance' is perhaps a misused word here.

Line 352 - 'was' to 'were'

Line 356 - Suggestion: 'One possible reason for this concerns the viable coal types' to 'One possible reason for this is the various coal types'

Line 427 - remove 'be'

---

## Author Comment (AC1) · 14 Nov 2019

The comment was uploaded in the form of a supplement:
https://www.atmos-chem-phys-discuss.net/acp-2019-584/acp-2019-584-AC1-
supplement.zip

---

## Author Comment (AC2) · 14 Nov 2019

The comment was uploaded in the form of a supplement:
https://www.atmos-chem-phys-discuss.net/acp-2019-584/acp-2019-584-AC2-supplement.zip

---

## Author Comment (AC3) · 15 Nov 2019

The comment was uploaded in the form of a supplement:
https://www.atmos-chem-phys-discuss.net/acp-2019-584/acp-2019-584-AC3-
supplement.zip

---

## Author Comment (AC4) · 15 Nov 2019

The comment was uploaded in the form of a supplement:
https://www.atmos-chem-phys-discuss.net/acp-2019-584/acp-2019-584-AC4-supplement.zip

---

## Author Response (AR1)

**Response to co-editor**

Journal: ACP

Title: "Molecular compositions and optical properties of dissolved brown carbon in smoke particles illuminated by excitation-emission matrix spectroscopy and Fourier-transform ion cyclotron resonance mass spectrometry (FT-ICR MS) analysis"

Author(s): Jiao Tang et al.

MS No.: acp-2019-584

Dear co-editor:

In compliance with your suggestions and the reviewers' detailed comments, we carefully revised the manuscript. We checked the text and references again.

We appreciate the two reviewers for their helpful comments and detailed language corrections on our manuscript. We considered the detailed comments by the reviewers and responded to their suggestions and questions. According to the comments on the mass spectra from two reviewers, we reconstructed this part and rewritten it. For your and reviewer's easiness to review the manuscript, an annotated manuscript was attached. As a result, the clarity of the manuscript should be greatly improved. We sincerely appreciate your consideration. Look forward to hearing from you soon.

With Best Regards,

Dr. Jun Li

State Key Laboratory of Organic Geochemistry

Chinese Academy of Sciences

**Comments from reviewers**

**Reviewer#1**

This manuscript describes measurements of the fluorescence of atmospheric WSOC, classifying it into separate types using parallel factor analysis, and attempting to correlate these types with high-resolution mass spectrometry data. The measurements of a very good technical quality and will be of interest to those who study aerosol fluorescence, organosulfates, or organonitrates. The work will also support future source apportionment of aerosol by fluorescence measurements. However, some interpretations, conclusions and assertions are not adequately supported by the data presented. For this reason, major revision of the discussion of the mass spectrometry results (and the second half of the abstract) is needed. The work is potentially publishable after addressing the comments below.

*Response: Thanks for your recognition of our work. Due to FT-ICR MS only can provide the molecular composition, not chemical structure. So, the possible structure we gave may be doubt just dependent on these existing data. According to the comments gave of you, we decided to delete the part of mass spectra and rewrote it.*

*We added the new parts in sections 3.3 and 3.4 (lines 402-705) in the revised manuscript, which mainly discussed the molecular composition in different solutions of various sources and determined their unique characteristics among these sources. Additionally, some modifications and discussions were added in the abstract, introduction, and conclusion in the revised manuscript.*

The citation of previous work in the manuscript has some gaps. For example, it is odd that the manuscript comments on similarities between HULIS and terrestrial humic

acids (line 70) without citing the authoritative review on this subject by Graber and Rudich.(1) One A. Laskin paper in the Results section is erroneously cited as "Alexander et al. 2009".

*Response: Many thanks for your suggestions. We have added the paper in the text and revised the erroneous cite.*

*Please see line 80 and line 409.*

The authors fundamentally assume that a correlation between a fluorescent component and a set of MS formulas (like "CHON") means that they are determining the molecular compositions of the fluorescent molecules. This may not be true. Molecules with high electrospray ionization efficiencies are not necessarily the same as those with high absorbance or fluorescence. It would therefore be fortuitous if major ESI ions were the same ones responsible for observed absorbance or fluorescence, without extensive and identical chromatographic separation before each technique.(2, 3) Furthermore, it has been shown that many non-polar brown carbon components in biomass burning aerosol cannot be ionized by ESI.(3) The manuscript should discuss these issues.

*Response: Thanks for your suggestions. Previous studies used the correlation analysis to speculate the chemical structures of fluorescent components (Chen et al., 2016a,b; Stubbins et al.,2014). Chen et al., (2016a,b) conducted the correlation analysis of the relative intensities of ion groups (including $C_x$, CH, $CHO_1$, CHN, $CHO_1N$, $CHO_{>1}N$, CS, CO, $HO,CO_2$, $C_2H_4O_2^+$) in the HR-AMS spectra and relative contents of the fluorescent components. They also found the relationships between relative contents of fluorescent components and relative intensities of chemical groups (C-H, $C-NH_2$, C-OH, $C-ONO_2$, C=O, COOH) in the FT-IR spectra. Stubbins et al., (2014) conducted a Spearman's correlations between relative intensities of FT-ICR MS peaks and the relative intensities of fluorescent components and speculated the possible structure of chromophores. These studies are the basis of our research. However, as you said, this*

*method may be inadequate without chromatographic separation. Thus, we rewrote this part of mass spectra.*

*The new revised parts were presented in section 3.3-3.4. We discussed the difference in molecular composition in the different sources and determined the factors that affected the light absorption capacity in the source-emission aerosols.*

*Reference:*

*Chen, Q., Ikemori, F., and Mochida, M.: Light Absorption and Excitation-Emission Fluorescence of Urban Organic Aerosol Components and Their Relationship to Chemical Structure, Environ. Sci. Technol., 50, 10859-10868, https://doi.org/10.1021/acs.est.6b02541, 2016.*

*Chen, Q., Miyazaki, Y., Kawamura, K., Matsumoto, K., Coburn, S., Volkamer, R., Iwamoto, Y., Kagami, S., Deng, Y., Ogawa, S., Ramasamy, S., Kato, S., Ida, A., Kajii, Y., and Mochida, M.: Characterization of Chromophoric Water-Soluble Organic Matter in Urban, Forest, and Marine Aerosols by HR-ToF-AMS Analysis and Excitation-Emission Matrix Spectroscopy, Environ. Sci. Technol., 50, 10351-10360, https://doi.org/10.1021/acs.est.6b01643, 2016.*

*Stubbins, A., Lapierre, J. F., Berggren, M., Prairie, Y. T., Dittmar, T., and del Giorgio, P. A.: What's in an EEM? Molecular signatures associated with dissolved organic fluorescence in boreal Canada, Environ. Sci. Technol., 48, 10598-10606, 10.1021/es502086e, 2014.*

The interpretation of functional groups from the ESI-MS data should be explained more clearly (line 376). The authors try four different methods (Tables S9 – S16), but it is difficult to understand the differences between them.

*Response: It is very sorry for making a mistake for explaining this Table 2 as Figure 2. The classification method as follows: The introduction of functional groups was referred to the study of Chen et al., (2016b) and the ionized properties of ESI-. Left of Table 2 showed the classified method by the functional groups (such as $CHO_1$, $CHO_{>1}$,*

*CHO$_{\leqslant 2}$N, and CHO$_{>2}$N and so on), and the right showed the method according to the Ven diagram (Patriarca et al., 2019) that made a partition based on their molecular distribution, such as lignin-like, protein-like, lipid-like, and carbohydrates and so on. They were the two classified methods we used. In addition, the potential BrC components could be identified based on the research of Lin et al., (2018) which excepted these compounds that had a low unsaturation degree. Thus, before classification, according to whether determine the potential BrC or not, we obtained four classified methods to build the relationship between fluorescence and molecular composition for further speculating the possible structure of chromophores. Table S9-S16 presented all the results of correlation analysis between the relative intensities of ion groups of FT-ICR MS and the relative abundances of fluorescent components. Finally, we chose the best method of which these functional groups exhibited a significant correlation with these fluorescent components.*

*In the new revising part of this manuscript, we attempted to build the relationship between the light absorption capacity of dissolved BrC with the corresponding structures (such as DBE, O/C, MW and so on). The results were presented in Section 3.4 (line 643-705).*

***Reference:***

*Chen, Q., Ikemori, F., and Mochida, M.: Light Absorption and Excitation-Emission Fluorescence of Urban Organic Aerosol Components and Their Relationship to Chemical Structure, Environ. Sci. Technol., 50, 10859-10868, https://doi.org/10.1021/acs.est.6b02541, 2016.*

*Patriarca, C., Bergquist, J., Sjoberg, P. J. R., Tranvik, L., and Hawkes, J. A.: Online HPLC-ESI-HRMS Method for the Analysis and Comparison of Different Dissolved Organic Matter Samples, Environ. Sci. Technol., 52, 2091-2099, https://doi.org/10.1021/acs.est.7b04508, 2018.*

The authors should explicitly describe and justify their assumptions in assigning functional groups to formulas. For example, it appears that all compounds with S:O ratios of 1:4 have been assigned as organosulfates, while higher ratios are assigned as organosulfates with additional oxygen functional groups, and lower ratios are assigned to sulfonates. The following chemical arguments suggest that these assignments are C2 not only arbitrary, but incorrect. Sulfonates form from S(IV) + carbonyl reactions, and these reactions also generate products with S:O ratios of 1:4, but with a C-S bond. Thus, this reviewer would argue that the authors' use of S:O ratios to distinguish between sulfonates and organosulfates is invalid. Furthermore, organosulfate production is thought to require acid catalysis at very low pH. The measured near-neutral pH of the WSOC extracts in this work suggests that acids have been mostly neutralized, and therefore organosulfate formation (via acid catalysis) appears to be unlikely. As another example, the assignment of C17H16O4 and C18H16O4 ions from the C2 group to "esters" could use further justification.

*Response: Thanks for your suggestion. When $S:O>4$, the $-OSO_3H$ structure (sulfate group) is more easily deprotonates by ESI-, and these compounds are more likely organosulfates, which has been demonstrated by many studies (Jiang et al., 2016, Mo et al.,2018; Song et al.,2018). When $S:O<4$, it is not possible to form the $-OSO_3H$ structure due to few O atoms, and It may form C-S bonds or sulfonates. These structures that presented here were speculated and not identified, because FT-ICR mass spectra could not provide structural information. In fact, having studies had detected similar chemical structure in biomass burning and coal combustion experiments which assigned as organosulfates (Song et al., 2018;). Maybe we provided a wrong structure for the assignment of $C_{17}H_{16}O_4$ and $C_{18}H_{16}O_4$ ions to "esters", because we think it cannot ionized by the ESI-.*

*Reference:*

*Mo, Y., Li, J., Jiang, B., Su, T., Geng, X., Liu, J., Jiang, H., Shen, C., Ding, P., Zhong, G., Cheng, Z., Liao, Y., Tian, C., Chen, Y., and Zhang, G.: Sources, compositions, and optical properties of humic-like substances in Beijing during the 2014 APEC summit: Results from dual carbon isotope and Fourier-transform ion cyclotron resonance mass*

*spectrometry analyses, Environ. Pollut., 239, 322-331, https://doi.org/10.1016/j.envpol.2018.04.041, 2018.*

*Song, J., Li, M., Jiang, B., Wei, S., Fan, X., and Peng, P.: Molecular Characterization of Water-Soluble Humic like Substances in Smoke Particles Emitted from Combustion of Biomass Materials and Coal Using Ultrahigh-Resolution Electrospray Ionization Fourier Transform Ion Cyclotron Resonance Mass Spectrometry, Environ. Sci. Technol., 52, 2575-2585, https://doi.org/10.1021/acs.est.7b06126, 2018.*

*Jiang, B., Kuang, B. Y., Liang, Y., Zhang, J., Huang, X. H. H., Xu, C., Yu, J. Z., and Shi, Q.: Molecular composition of urban organic aerosols on clear and hazy days in Beijing: a comparative study using FT-ICR MS, Environ. Chem., 13, 888-901, https://doi.org/10.1071/en15230, 2016.*

Line 63: The phrase "little structural information is available" is misleading. There have been several studies, most involving Alex Laskin, that determined detailed chemical structures in biomass burning aerosol extracts. Some of these studies are eventually cited in this manuscript, but they should be briefly summarized here.

**Response:** *Thanks for your suggestions. The HPLC-PDA-HRMS has been used to investigate optical properties and chemical composition of BrC compounds. Thus, we have revised it.*

*Please see lines 67-68 in the revised manuscript. We revised it as the following phrase "Recently, many studies have investigated the optical properties and molecular characteristic of BrC in laboratory simulated combustion (Budisulistiorini et al., 2017;Lin et al., 2018;Lin et al., 2016;Song et al., 2019) and their light absorption in controlled vehicle emissions (Xie et al., 2017)".*

Line 103: It is inappropriate to follow the statement "Concerns about the environmental and health effects of vehicle emissions have existed for decades" with only a single citation from 2015, unless the citation is a comprehensive review. This underreferencing happens at several points in the introduction.

*Response: Thanks for your suggestions. We have revised and checked it.*

*The resived parts were in lines 111-114. The revised phrase is "Residential CC and BB emissions, and motor vehicle emissions are significant anthropogenic sources of air pollutants, exceptionally fine particulate matter (PM $_{2.5}$) on urban and regional scales (Gentner et al., 2017;Yan et al., 2015;Zhang et al., 2018;Chen et al., 2015)".*

Line 278: It is stated that measured MAEs are higher in this study than in previous lab studies of biomass and coal burning aerosol. Some explanation for this difference should be given. There is no other discussion of MAEs in the manuscript.

*Response: Thanks for your suggestions. The topic on our study is the fluorescence characteristic from different sources and dig out the chemical structures of chromophores using FT-ICR MS. Therefore, that is the reason we explained little about the light absorption in this study. According to your suggestions, we added more discussions about the light absorption in different sources, and conducted a comparison of their light absorption.*

*Please see lines 270-310 in the revised manuscript.*

Line 320: The meaning of this sentence is unclear. What "other" fluorescent components are referred to?

*Response: The "other" denoted the remaining fluorescent components except for C3 and C1.*

*In the revised manuscript, we deleted this sentence because we found it is useless.*

Lines 33, 413, 464, 514: The authors appear to treat negative and positive correlations the same way in their interpretations. If P1 and P6 are negatively correlated with HCHOS, this would mean that P1 and P6 fluorophores are formed whenever H-CHOS is not present, but the authors go on to attribute P1 and P6 to H-CHOS here, in the

conclusion, and in the abstract.

*Response: Thanks for your suggestion. We have rewritten this part, please see Section 3.3-3.4.*

Line 35, 519-520: These conclusions are questionable. See earlier comments about esters and sulfonates versus organosulfates

*Response: The structures of these fluorescent components were speculated and not identified. Thus, we deleted this part and rewrote it.*

*Please see Section 3.3-3.4 in the revised manuscript.*

Tables S5 and S6: It seems inappropriate to report either averages or total intensities across different types of samples like this, without some justification.

*Response: According to your suggestion, we have deleted it.*

Technical corrections

There are a fair number of grammatical errors in the manuscript, which are not listed here. Fortunately, the meaning usually remains clear.

*Response: Thanks, we have carefully checked it.*

Line 49: the authors refer to the "near UV and UV/visible ranges," which are overlapping. Do they mean "near UV and visible ranges"?

*Response: Thanks for your reminder, we have corrected it.*

*We revised "near UV and UV/visible ranges," to "near UV and visible ranges" in the revised manuscript. Please see line 57.*

Line 162: this statement would be clearer if the difference was explained. Is it true that MSOC is different in this study because WSOC has already been removed?

*Response: As we stated in the text, this fraction was that we used methanol to re-extract the water-extracted filter. So it is true that MSOC is different from WSOC.*

*In the revised manuscript, we explained the reason we used the methanol to secondary-extraction. Please see lines 167-171.*

Line 213: The authors should briefly discuss what kinds of compounds will be missed by negative mode ESI. Will N-heterocycles be detected?

*Response: Thanks for your suggestion. The missing compounds in the ESI- may be some nonpolar or less polar compounds such as PAHs, O-PAHs, N-PAHs, and saturated hydrocarbons.*

*We added this discuss in the revised manuscript. Please see line 404-408.*

Line 241: "componet" should be "component"

*Response: Thanks, we have revised it in the revised manuscript.*

Line 366: What kinds of differences? Differences in fluorescence? The fluorescence of sample 36 is not shown in Figures S9 or S10.

*Response: The differences denoted the relative abundance of CHON in anthracite combustion and bituminous coal combustion. The previous study indicated that CHON was mainly detected in the biomass burning particles, but our study found that the CHON was abundant in coal combustion particles. Thus, we introduced the types of coal fuels that may be responsible for these differences. The sample 36 should be 38.*

*In the revised manuscript, we rediscussed it. Please see lines 493-495.*

Line 371: This description of axes appears to be describing Figures 3 and 4, but these figures are not mentioned until later

*Response: The axes were not used to describe Figure 3 and 4, but only used to determine the potential BrC compounds where data inside the region were potential BrC.*

Line 375: The wrong figure is referenced here, I think

*Response: Thanks. This should be Table 2.*

Figures S2 and S5: "resident" should be "residual"

*Response: Thanks. We have revised it.*

Figures S9 and S10: It would save readers a lot of time searching around if these samples where labeled with their sources on this graph (e.g. "bituminous coal")

*Response: According to your suggestions. In the revised version, we labeled the sources on the graph.*

*Please see Figure 4, 6 in the text and Figure S9, S10, S11, S18 in the revised manuscript.*

Figure S11 caption: Different regions are identified, but not labeled on the graph. It is not clear what the reader can do with this information without further graphing.

*Response: Thanks. According to your suggestions. In the revised manuscript, we marked this information on the graph.*

References Cited: 1. Graber, E. R.; Rudich, Y., Atmospheric HULIS: How humiclike are they? A comprehensive and critical review. Atmos. Chem. Phys. 2006, 6, 729-753. 2. Lin, P.; Aiona, P. K.; Li, Y.; Shiraiwa, M.; Laskin, J.; Nizkorodov, S. A.; Laskin, A., Molecular Characterization of Brown Carbon in Biomass Burning Aerosol Particles. Environ Sci Technol 2016, 50, (21), 11815-11824. 3. Lin, P.; Fleming, L. T.; Nizkorodov, S. A.; Laskin, J.; Laskin, A., Comprehensive Molecular Characterization of Atmospheric Brown Carbon by High Resolution Mass Spectrometry with Electrospray and Atmospheric Pressure Photoionization. Anal. Chem. 2018, 90, (21), 12493- 12502.

*Response: We had cited the references of 2 and 3 in the previous version. In the revised version, we add the new cite in the line 80.*

**Reviewer# 2**

This manuscript by Tang et al. describes a detailed chemical analysis on atmospheric brown carbon (BrC) extracted from smoke particles samples. Particle samples were collected from biomass burning, coal combustion, and vehicular emissions. Filter samples were extracted by either water or methanol and were analyzed with emission excitation matrix (EEM) and FTICR-MS with ESI(-) ionization. Six components were extracted from the EEM data using a parallel factor analysis method. A significant amount of effort was present to make correlations between these EEM components with functional groups determined with FTICR-MS. The authors concluded that correlations were observed between EEM components and certain functional groups, indicating that this method can be useful in source apportionment of BrC.

The topic of the manuscript is in-line with the scope of ACP, in particular, the importance of BrC in the atmosphere is emergent, but there is extremely limited chemical information on important individual chromophores. The manuscript is attempting to address this important question. However, I do not recommend publication in ACP in the current form. In addition to a few major scientific questions, I have significant concerns regarding the literary presentation of the manuscript. It requires a substantial refinement before it can be published in any journal. In particular, I found the manuscript very difficult to read due to ill-structured order of discussion, missing or repeated explanations for abbreviations, frequent references to the SI, as well as numerous grammatical and typological errors.

*Response: Thanks. According to your suggestions. We made a major revision and rewrote the part of FT-ICR mass spectra. We added the analysis of chemical information in different sources and discussed the differences among these sources. We revised the wrong parts, reconstructed the structure of discussion in the revised*

*manuscript.*

Major comments:

EEM and ESI (-) are powerful analytical methods, but I'm afraid that they are not quantitative enough to make meaningful correlation analysis. Light absorptivity should be the primary concern for BrC chromophores, but fluorescence intensity, which is the core of the analysis here, depends on a number of other factors. Meanwhile, ESI(-) is particularly sensitive to compounds with acidic hydrogens, but not to PAHs and other compounds unless they have a carboxylic group. I'm afraid that the positive correlation could be driven by the detection sensitivities of the two methods.

*Response: Previous studies used the correlation analysis to speculate the chemical structures of fluorescent components (Chen et al., 2016a,b; Stubbins et al.,2014). These studies were the basis of my research. It is true because ESI- can only ionize polar compounds and will ignore other strongly absorbing compounds, such as O-PAHs, N-PAHs. According to your suggestions, we thought that using the FT-ICR MS to speculate the possible structures and provide a fundamental confirmation may be very difficult without further chromatographic separation. Thus, we retreated our data and rewrote the part.*

*The reconstructed part were mainly presented in Section 3.3-3.4. Also, we updated the results and made some modifications in abstract, introduction, and conclusions.*

*Reference:*

*Chen, Q., Ikemori, F., and Mochida, M.: Light Absorption and Excitation-Emission Fluorescence of Urban Organic Aerosol Components and Their Relationship to Chemical Structure, Environ. Sci. Technol., 50, 10859-10868, https://doi.org/10.1021/acs.est.6b02541, 2016.*

*Chen, Q., Miyazaki, Y., Kawamura, K., Matsumoto, K., Coburn, S., Volkamer, R., Iwamoto, Y., Kagami, S., Deng, Y., Ogawa, S., Ramasamy, S., Kato, S., Ida, A., Kajii, Y., and Mochida, M.: Characterization of Chromophoric Water-Soluble Organic*

*Matter in Urban, Forest, and Marine Aerosols by HR-ToF-AMS Analysis and Excitation-Emission Matrix Spectroscopy, Environ. Sci. Technol., 50, 10351-10360, https://doi.org/10.1021/acs.est.6b01643, 2016.*

*Stubbins, A., Lapierre, J. F., Berggren, M., Prairie, Y. T., Dittmar, T., and del Giorgio, P. A.: What's in an EEM? Molecular signatures associated with dissolved organic fluorescence in boreal Canada, Environ. Sci. Technol., 48, 10598-10606, 10.1021/es502086e, 2014.*

The authors use the FTICR-MS to rule out functional groups. Although the authors present a thoughtful interpretation of the FTICR-MS data, caution is required, as what MS provides is the elemental composition, not functional group information. For example, the chemical structures shown in Figure 5 do not contain any acidic functional group, and I double if they can be detected by ESI (-).

*Response: The classified method of function groups was based on the Chen et al. 2016b and the ionized properties of ESI-. In the results of Chen et al., they found fluorescent components had a good correlation with these chemical groups (C-H, C-$NH_2$, C-OH, C-$ONO_2$, C=O, COOH), and with ion groups ( $C_x$, CH, $CHO_1$, CHN, $CHO_1N$, $CHO_{>1}N$, CS, CO, $HO,CO_2$, $C_2H_4O_2^+$) of HR-AMS. Thus, we tried to build the relationship between fluorescent components and ion groups of FT-ICR MS. The chemical structures provided in Figure 5 were speculated, not determined.*

*In the revised manuscript, we deleted this part, and rewrote it (Section 3.4).*

*Reference:*

*Chen, Q., Ikemori, F., and Mochida, M.: Light Absorption and Excitation-Emission Fluorescence of Urban Organic Aerosol Components and Their Relationship to Chemical Structure, Environ. Sci. Technol., 50, 10859-10868, https://doi.org/10.1021/acs.est.6b02541, 2016.*

Table 2 is a critical part of the manuscript, presenting the functional group assignment based on FTICR-MS data. However, no explanation is provided for the table at all in the manuscript. Is the left side of the table linked to the right side of the table? The categories shown on the right side of Table 2 (Lipids, proteins, etc.) seem very irrelevant to atmospheric particles, but no explanation is provided in the main text.

*Response: Sorry for making this mistake. The Figure 2 should be Table 2 (Line 375). The left side and right side of the table were the two methods we classified to explore the possible structures of fluorescent components. The right side of the Table 2 is the categories that the compounds located in the domains in Ven krevelen (VK) diagram. In addition, Vk diagram can provide a visual graphic display of compound distribution and identify different composition domains in organic mixtures.*

*This part in the revised manuscript has been deleted.*

*Reference:*

*Patriarca, C., Bergquist, J., Sjoberg, P. J. R., Tranvik, L., and Hawkes, J. A.: Online HPLC-ESI-HRMS Method for the Analysis and Comparison of Different Dissolved Organic Matter Samples, Environ. Sci. Technol., 52, 2091-2099, https://doi.org/10.1021/acs.est.7b04508, 2018.*

The authors have presented a huge amount of work in interpreting the EEM components, FTICR-MS data, as well as the correlation analysis. The authors deserve a lot of credit for doing such a full-bodied analysis. However, the current conclusions in the manuscript do not appear very helpful for the atmospheric chemistry community, other than demonstrating the heterogeneity and complexity of the system. The authors should reconstruct the discussion and conclusion with more atmospheric implications.

*Response: In fact, EEMs-based method to characterize the atmospheric BrC have been widely used. However, there is no classification system of chromophores in the atmosphere by using fluorescence method, mainly referring to the aquatic environment. Due to the differences in biochemical behavior, transport process, and formation*

*mechanism, there is a big gap between atmospheric and aquatic environment. Therefore, the fluorescence spectra of different sources may provide certain chromophores linked with sources, and combined with FT-ICR MS to speculate the possible structures of fluorescent components, which would provide a reference fluorescence spectrum for the study of brown carbon in the atmosphere.*

*In the revised manuscript, we deleted this part and reconstructed the discussions and conclusions.*

Minor Comments

The manuscript is titled as "BrC in smoke particles". I personally felt odd that vehicular emissions are also included as smoke particles.

*Response: Thanks. According to your suggestion, we have revised the title.*

*The revised title is "Molecular compositions and optical properties of dissolved brown carbon in biomass burning, coal combustion, vehicle emission aerosols illuminated by excitation-emission matrix spectroscopy and FT-ICR MS analysis".*

I am not a specialist in PARAFAC and found it difficult to see the concepts and purpose of PARAFAC until the end of Section 2.5. I recommend the authors added an introductory statement for PARAFRAC either in Introduction of Section 2.5. For example: "The purpose of PARAFRAC is to extract X components from the EEM data based on . .

*Response: Thanks. According to your suggestion, we added the concept of PARAFAC in the introduction.*

*Please see lines 88-90 in the revised manuscript.*

Regarding water vs methanol extraction. The objectives of investigating WSOC and MSOC is unclear. Is the purpose to investigate BrC with distinct polarities? Is it to investigate "fat-soluble" fraction (Line 271)?

*Response: Previous studies indicated water could not effectively extract the BrC in the aerosols, and found the water-insoluble organic fraction had higher light absorption capacity than water-soluble organic fraction (Chen et al., 2017). So we want to know more about the water-insoluble BrC.*

*In the revised manuscript, we added an explanation. Lines 167-170.*

*Reference:*

*Chen, Q., Ikemori, F., Nakamura, Y., Vodicka, P., Kawamura, K., and Mochida, M.: Structural and Light-Absorption Characteristics of Complex Water-Insoluble Organic Mixtures in Urban Submicrometer Aerosols, Environ. Sci. Technol., 51, 8293-8303, https://doi.org/10.1021/acs.est.7b01630., 2017.*

Related to the previous point, a discussion is needed on why the WSOC and MSOC are so distinct. To my understanding, these two solvents should extract different, but somewhat overlapping classes of organic compounds.

*Response: Water extracts high-polar compounds and methanol extracts medium-polar compounds. If organic matters were extracted individually by water and methanol, there were certainly overlapping compounds in the two fractions. However, when we extracted the organic matters using water, and then using methanol, there were fewer overlapping compounds. Meanwhile, we will know more about the variation of optical properties and molecular compositions of water-insoluble organic matter comparing with water-soluble organic matter.*

Line 266 - It is a little confusing because Figure 2b is introduced before Figure 1 and the PARAFRAC components. Can the authors consider making Figure 2b an individual figure? Also, to make an argument on MAE is 'higher' or 'lower', more statistics are

needed. Instead of presenting Figure 2b as is, I would recommend using a more statistical approach, such as a box and whisker plot.

*Response: Thanks for your suggestions. We have replotted the figure and discussed the light absorption capacity among these different sources, and made a comparison with the other studies.*

*Please see line 270-311 in the revised manuscript and the revised Figure 1.*

Line 292 - What is 'region IV'?

*Response: The "region IV' (Ex=225-250nm, Em=356-400nm) is defined as protein-like or tryptophan-like fluorophore according to the previous study (Qin et al., 2018).*

*Reference:*

*Qin, J., Zhang, L., Zhou, X., Duan, J., Mu, S., Xiao, K., Hu, J., and Tan, J.: Fluorescence fingerprinting properties for exploring water-soluble organic compounds in PM 2.5 in an industrial city of northwest China, Atmos. Environ., 184, 203-211, https://doi.org/10.1016/j.atmosenv.2018.04.049, 2018.*

Paragraph starting Line 369. It is very confusing that the paragraph started with an introduction to DBE, but the topic rapidly changed to O/C and H/C. The authors should consider reordering the discussion here.

*Response: In this sentence, the introduction of DBE is to determine the potential BrC using the method and then classified the ion groups.*

*In the revised manuscript, we had rewritten it.*

What is AImod?

*Response: $AI_{mod}$ is a modified AI value, which can be calculated by considering only half of the oxygen being present in carbonyl functional groups (Koch et al., 2006). AI*

*is the most conservation case and result in underestimation of the structures. Although AImod may improve the aromatic index, it may introduce uncertainties.*

Figure 1 - no color scale explanation. Is each graph normalized to its highest intensity? The readers cannot see the relative importance of the 6 components (i.e., are one or two components much more intense than others?)

*Response: The fluorescence did not normalize by its highest intensity, but the Raman peak of water. In the revised manuscript, we added the score of the 6 components.*

Figures 4 and 5- the authors introduced a region between slope 0.5 and 0.9 on the DBC vs C plot (Line 370). Why not show these lines in Figure 4 and 5?

*Response: According to your suggestion, we added the two linear in the relative graphs in the revised manuscript. (Figure 6; Figure S15, S17, S18 ).*

Technical Comments - there are more grammatical errors than listed here, please check. Line 39 - the abbreviation of EEM is already introduced in Line 23.

*Response: Thanks. We have checked these errors carefully in the revised manuscript.*

Line 82 and Line 85 - Chen et al / Lee et al are repeated

*Response: Thanks, We have revised it in the revised manuscript.*

*Please see line 90, Chen et al., (2016b) observed…*

*Line 93, Lee et al., (2013) illustrated…*

Line 110 - the abbreviation of EEM is already introduced in Line 64.

*Response: Thanks for reminder, we have revised it and use the abbreviation..*

Line 138 -'difficult' to 'difficulty'

*Response: Thanks for the reminder. We have revised it.*

Line 141 - 'Every coal about 1 kg fuels was burned three times'. To 'Coal (~ 1 kg each) was burned in triplicate."

*Response: The correct expression is that "every coal was burning three times, about 1kg fuels per burn". And we have revised it.*

Line 144 - 'Additional' to 'Additionally,'

*Response: Thanks. We have revised it.*

Line 151 - 'truck' to ' a truck' or 'trucks

*Response: Thanks. We have revised it.*

Line 162 - 'MSOC fraction from the methanol extract' is redundant.

*Response: Thanks. We have deleted it.*

*We revised the 'MSOC fraction from the methanol extract' to 'MSOC' in line 170 in the revised manuscript.*

Line 164 - 'um'

*Response: Thanks. We have revised it.*

*We have changed 'um' to 'μm' in line 172 in the revised manuscript.*

Line 237 - 'Additional' to 'Additionally,'

*Response: Thanks. We have revised it.*

*We have changed the 'Additional' to 'Additionally,' in line 246 in the revised manuscript.*

Line 274 to 277 - MAE was higher in methanol extract for biomass burning and coal

samples. I could not follow why that indicates a greater variation in chemical composition in MSOC.

*Response: Thanks. It may be some wrong express. According to your suggestion, we have deleted it.*

Line 352 - 'abundance' is perhaps a misused word here

*Response: Thanks for your reminder. We have revised it.*

*We changed 'anundance' to 'abundant' in line 390 in the revised manuscript.*

Line 352 - 'was' to 'were'

*Response: Thanks. We have revised it.*

Line 356 - Suggestion: 'One possible reason for this concerns the viable coal types' to 'One possible reason for this is the various coal types'

*Response: Thanks for your suggestion. When we rewrote this part, some relative errors may have been deleted in the revised manuscript.*

Line 427 - remove 'be'

*Response: Thanks for your suggestion. Because we rewrote this part, some relative errors may have been deleted in the revised manuscript.*

---

## Referee Report (RR1)

The manuscript appears to be very strong after the revision, and I recommend publication in ACP after a few technical revision.

Technical comments (there may be more, please proofread):

Line 31- "undefinition" to "undefined"
Line 199 - "thought" to "though"
Line 275 - "at there" to "there"
Line 325 - "almost" to "mostly"?
Line 325 - "region IV" was never introduced in the main text. If it is necessary to be named as region IV, the authors should define it. If not necessary, the authors can consider removing the naming of region IV and just mention that it is a region categorized as protein -like…
Line 345 - "indicating different chemical structures". I understood that the authors performed water extraction first, then methanol extraction. I just wanted to suggest that this is a good location to briefly remind the readers about this by discussing that the water-soluble fraction is already extracted with the WSOC, and MSOC contains a distinct population of compounds.
Line 396-397 - "Combing these results with the WSOC mentioned above results and comparing the…" Awkward sentence; please restructure.
Line 408 - "easily ioninzed" to "readily ionizable"
Line 466 - "had a resemble VK diagram to that of…" to "had a VK diagram similar to that of.."
Line 498 - "new sight for" to "new insight into"?
Line 559 - "could be more detected in the atmosphere" - unclear phrase.
Line 627 - "Comparison with" to "Compared to"
Line 628 - "common in" should be "common to" here
Line 669 - "Next, we further discussed" - did the authors mean "In the next section, we will further discuss"?
Line 671 to 673. This sentence was very unclear

---

## Author Response (AR2)

**Response to co-editor**

**Journal: ACP**

Title: "Molecular compositions and optical properties of dissolved brown carbon in smoke particles illuminated by excitation-emission matrix spectroscopy and Fouriertransform ion cyclotron resonance mass spectrometry (FT-ICR MS) analysis" Author(s): Jiao Tang et al.

MS No.: acp-2019-584

**Dear co-editor:**

We are pleased for our work to be accepted on ACP. In compliance with your suggestions and the reviewers' detailed comments, we carefully revised the manuscript. We adjusted the Figure and modified the size of fonts in the Figure for clearer. We checked the text and references again. We responsed to their suggestions and questions.

With Best Regards,

Dr. Jun Li State Key Laboratory of Organic Geochemistry Chinese Academy of Sciences

**Reviewer#1**

Generally beginning sentences with "Besides, ..." should be avoided.

Response: Thanks for your suggestion. We have checked the manuscript to avoid such sentence patterns.

Line 97: the meaning of this sentence is unclear.

Response: Thanks for your suggestion. This sentence has been revised as follows: However, when analyzing chromophoric BrC using fluorescence spectra, the challenges are the lack of a classification system for fluorescence components, to distinguish chromophores from most non-absorbing constituents and to determine the chemical structures of the chromophores.

Line 136: "end until" should be "ended when" Response: Thanks for your comment. We have revised it.

Line 159: the meaning of this sentence is unclear.

Response: In this study, the vehicle emission aerosols contained two parts: tunnel aerosols samples (more aging aerosols) and vehicle exhaust particles (more fresh aerosols). When they have similar chemical properties or optical properties, we used vehicle emissions to represent them. Otherwise the opposite. In the revising manuscript, we have modified this sentence as follows: With no other instructions, we used "vehicle emissions" to represent all tunnel aerosols and vehicle exhaust particle samples.

Line 275: the meaning of the sentence beginning with "At there," is unclear. Response: Thanks for your hint. We have changed it to "Here".

Line 499: the meaning of this sentence is unclear.

Response: Thanks for your hint. We have deleted this sentence because this result cannot be obtained by limited data without further validation.

Line 552: "elementals" should be "elements"?

Response: Thanks for your hint. We have corrected it in the revised manuscript.

Line 589: "molecular" should be "molecules"? "located" should be "are located" at two places in this sentence.

Response: Thanks for your suggestion. We have made a revision in the revised manuscript.

Figure 5 uses both "vehicle emissions" and "vehicle exhaust", but vehicle emission MSOC looks like tunnel WSOC and not vehicle exhaust WSOC. This is confusing. Response: Thanks for your suggestion. We have revised it in Figure 5 in the revised manuscript. We used vehicle emission to represent both tunnel sample and vehicle exhaust.

Line 677: "well positive" should be "significant positive" Response: Thanks for your suggestion. We have revised it in the revised manuscript.

Line 705: "O-PAH" should be "N-PAH"

Response: Thanks for your suggestion. We have corrected it in the revised manuscript.

**Reviewer#2**

The manuscript appears to be very strong after the revision, and I recommend publication in ACP after a few technical revision.

Technical comments (there may be more, please proofread):

Line 31- "undefinition" to "undefined"

Response: Thanks for your comment. We have revised it in the revisedmanuscript.

Line 199 - "thought" to "though"

Response: Thanks for your reminder. We carefully checked it and found it is more appropriate to change "generally thought it did not affect the absorbance according to the prior study" to the following sentence: which generally did not affect the absorbance according to prior study.

Line 275 - "at there" to "there"

Response: Thanks for your reminder. We thought to change "at there" to "here", which would be more appropriate.

Line 325 - "almost" to "mostly"?

Response: Thanks for your suggestion. We have corrected it in the revised manuscript.

Line 325 - "region IV" was never introduced in the main text. If it is necessary to be named as region IV, the authors should define it. If not necessary, the authors can consider removing the naming of region IV and just mention that it is a region categorized as protein -like...

Response: Thanks for your suggestion. We decided to remove the naming of region IV and revised as follows: "in the region categorized as protein-like (cytidine) or tryptophan-like fluorophore."

Line 345 - "indicating different chemical structures". I understood that the authors performed water extraction first, then methanol extraction. I just wanted to suggest that this is a good location to briefly remind the readers about this by discussing that the water-soluble fraction is already extracted with the WSOC, and MSOC contains a distinct population of compounds.

Response: Thanks for your suggestion. We have changed "indicating different chemical structures" to "indicating MSOC contained different compound types from WSOC after water extraction" for better understanding for the readers.

Line 396-397 - "Combing these results with the WSOC mentioned above results and comparing the..." Awkward sentence; please restructure.

Response: Thanks for your suggestion. We have revised it as follows "In summary, the variation of the fluorescent components from different sources obtained by EEM-PARAFAC method could be helpful to the source apportionment of BrC in environment applications.

Line 408 - "easily ioninzed" to "readily ionizable"

Response: Thanks for your suggestion. We have corrected it in the revised manuscript.

Line 466 - "had a resemble VK diagram to that of…" to "had a VK diagram similar to that of.."

Response: Thanks for your suggestion. We have corrected it.

Line 498 - "new sight for" to "new insight into"?

Response: Thanks for your reminder. We have deleted this sentence because this result cannot be obtained by limited data without further validation.

Line 559 - "could be more detected in the atmosphere" - unclear phrase. Response: Thanks for your reminder. We have deleted it because it would be confused.

Line 627 - "Comparison with" to "Compared to" Response: Thanks for your suggestion. We have corrected it.

Line 628 - "common in" should be "common to" here Response: Thanks for your suggestion. We have corrected it.

Line 669 - "Next, we further discussed" - did the authors mean "In the next section, we will further discuss"?

Response: Yes. The following two paragraphs discussed the relationships. We have added "below" at the end of this sentence for clear expression.

**Line 671 to 673. This sentence was very unclear**

Response: Thanks for your suggestion. We have revised it as follows: Before discussing their relationships, we firstly determined these compounds that were potential to absorb light radiation based on the above statement to reduce the influence of non-absorbing substances (Lin et al., 2018)".

Molecular compositions and optical properties of dissolved brown carbon in
 biomass burning, coal combustion, vehicle emission aerosols illuminated by
 excitation-emission matrix spectroscopy and FT-ICR MS analysis

4 Jiao Tang1,4, Jun Li\*,1, Tao Su1,4, Yong Han2, Yangzhi Mo1, Hongxing, Jiang1,4, Min

5 Cui2.a, Bin Jiang1, Yingjun Chen2, Jianhui Tang3, Jianzhong Song1, Ping'an Peng1,

6 Gan Zhang \*,1

[revised manuscript text omitted]
 0.30 \text{ m}^2 \text{ g}^{-1}\text{C}$ ) but higher than MSOC ( $0.26 \pm 0.09 \text{ m}^2 \text{ g}^{-1}\text{C}$ ) in this study.